# RADAR: REASONING-ABILITY AND DIFFICULTY-AWARE ROUTING FOR REASONING LLMs

**Nigel Fernandez**[*1], **Branislav Kveton**[2], **Ryan A. Rossi**[2], **Andrew S. Lan**[1], **Zichao Wang**[2]
[1]University of Massachusetts Amherst,  [2]Adobe Research
{nigel,andrewlan}@cs.umass.edu, {kveton,ryrossi,jackwa}@adobe.com

## ABSTRACT

Reasoning language models have demonstrated remarkable performance on many challenging tasks in math, science, and coding. Choosing the right reasoning model for practical deployment involves a performance-cost trade-off at two key levels: model size and reasoning budget, where larger models and higher reasoning budgets lead to better performance but incur greater cost and latency. In this work, we tackle this tradeoff from the angle of model configuration routing for different queries, and present RADAR (**R**easoning–**A**bility and **D**ifficulty-**A**ware **R**outing), a lightweight, interpretable, and scalable routing framework. Inspired by psychometrics, RADAR learns an *item response model* from model responses with different budgets to different queries, with *interpretable* parameters including *query difficulties* and *model-budget abilities*. RADAR then routes queries with higher difficulty to model-budget pairs with higher ability, and vice versa. We conduct extensive experiments on 8 widely used challenging reasoning benchmarks, demonstrating the superior performance of RADAR compared to state-of-the-art model routing methods. RADAR also exhibits *query generalization* capabilities, achieving strong performance on out-of-distribution queries on all benchmarks. RADAR is also *scalable* and can efficiently integrate additional models by *dynamically selecting* a small set of evaluation queries to estimate their abilities.

## 1 INTRODUCTION

Recent advances in large language models (LLMs) have leveraged reinforcement learning (RL) (Shao et al., 2024) to train models to reason using chain-of-thought before generating an output. These reasoning language models (RLMs) (Yang et al., 2025; Guo et al., 2025; OpenAI & et al., 2024) have demonstrated impressive performance across a diverse range of challenging tasks, including math (MAA, 2024), science (Rein et al., 2024), coding (Jimenez et al., 2024), visual perception (Lu et al., 2024), and tool use (Yao et al., 2025). The excitement has led to a flurry of new open-source and proprietary RLMs; for example, Hugging Face already lists $4,492$ RLMs as of March 8th, 2026. These models have varying sizes, specialize in different domains, and offer various configurations, including reasoning budgets to balance performance and cost. For example, OpenAI's reasoning models (OpenAI & et al., 2024) have "low", "medium", and "high" reasoning budgets for developers to choose from depending on their application.

Always choosing the "best" and most expensive RLM configuration with the highest level of reasoning budget is not always the "right" choice for every query: for some simpler queries, there might exist a "worse" and cheaper RLM configuration with low or no reasoning budget that correctly answers the query, resulting in significant cost savings without sacrificing performance. Indeed, we empirically observe the same phenomenon in Figure 1, where we show that over $50\%$ of the queries from MATH-500 (Hendrycks et al., 2021c) can be solved using an RLM as small as Qwen3-0.6B with minimal reasoning budget (measured by the number of reasoning tokens). On the contrary, some challenging queries require a much more capable RLM with a high reasoning budget. Strong RLMs can also "over-think" which could hurt performance even for simple queries (Su et al., 2025; Hassid et al., 2025; Hong et al., 2025; Shojaee et al., 2025; Ghosal et al., 2025). This performance-cost tradeoff presents a challenge for practitioners: how to choose the "right" RLM and its configu-

---

*Work done during an internship at Adobe Research.

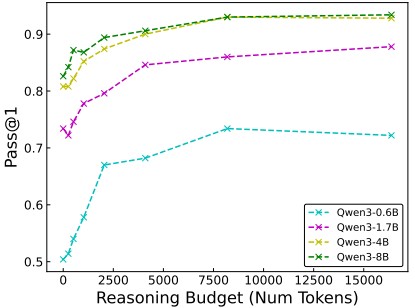 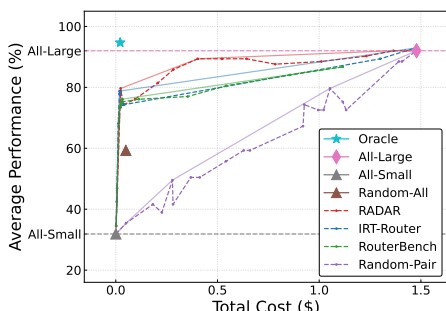

Figure 1: **Left**: Our pilot study on MATH-500 (Hendrycks et al., 2021c) shows performance difference over (RLM, reasoning budget) configurations with the smallest RLM already solving over 50% of the queries with minimal reasoning. **Right**: RADAR exploits this performance differential by jointly estimating query difficulties and configuration abilities, and routing queries to sufficiently able configurations, thereby optimizing performance-cost tradeoffs towards the Pareto front. On out-of-domain queries from FRAMES (Krishna et al., 2024), RADAR can match 90% of the performance of OpenAI o4-mini with a high reasoning budget at just 10% of its cost, with the next best method (Song et al., 2025) requiring 30% of the cost. Solid lines in the figure denote the convex hull corresponding to each curve.

ration (e.g., the reasoning budget) that is sufficiently capable of correctly answering a query, thereby maximizing performance while minimizing cost?

In this work, we propose a framework entitled RADAR (**R**easoning-**A**bility and **D**ifficulty-**A**ware **R**outing) to address the above challenge. Given a pool of {RLM, reasoning budget} configurations, a user-desired performance-cost tradeoff profile, and a new query, RADAR chooses the optimal configuration for the query according to the tradeoff profile. RADAR is lightweight and efficient: it decides the configuration in real time (∼7 milliseconds latency overhead) at the query level (it performs the assignment before the RLM ingests the query) and does not require model switching during generation, thus avoiding the need to re-query the RLM multiple times or recompute the KV-cache that could occur for cascading-based routers (Chen et al., 2023; Zhang et al., 2024). RADAR is also designed to be plug-and-play: it treats RLMs as black boxes and uses them as-is without the need to fine-tune them, which is convenient for practitioners who can use RLMs and their configurations in a standard API call. When a new RLM becomes available, RADAR can rapidly include it into its pool of {RLM, reasoning budget} configurations available for future queries.

The key enabling ingredient in RADAR is a custom item response theory (IRT) model, a classic technique inspired by psychometrics and educational assessment (Rasch, 1960; Lord, 2012; van der Linden & Hambleton, 1997; DeMars, 2010). We use an IRT model to jointly estimate *interpretable* query *difficulties* and RLM *abilities* at different reasoning budgets. Specifically, we first perform a calibration step, where we collect evaluation responses of each {RLM, reasoning budget} configuration to a collection of queries. We then model this evaluation response matrix via IRT to estimate the latent ability and difficulty parameters. To make this approach generalizable to out-of-distribution (OOD) queries, we parametrize the query difficulty using a learnable vector that, when multiplied by the query embedding obtained from an off-the-shelf embedding model, yields the query difficulty. We also parametrize each RLM configuration with a learnable, scalar-valued ability. To include a new RLM configuration in RADAR, we estimate its ability by evaluating it on a small set of dynamically selected queries, employing a classic technique inspired by adaptive testing in educational assessment (Wainer et al., 2000; Hofmann et al., 2025). These design choices enable RADAR to (1) handle new queries in real time and (2) generalize well to new RLM configurations.

We formulate model configuration selection as multi-objective optimization (MOO) that searches for the configuration at the Pareto front of the performance-cost tradeoff curve using scalarization techniques (Miettinen, 1999). MOO (Keeney & Raiffa, 1993; Emmerich & Deutz, 2018) is a well-established framework for optimizing multiple objectives, with major applications in engineering (Marler & Arora, 2004), product design and manufacturing (Wang et al., 2011), and economics (Ponsich et al., 2013). This work is the first application of MOO, beyond linear scalarization, to LLM routing. We conduct extensive experiments on 8 widely recognized challenging reasoning benchmarks. RADAR demonstrates superior performance compared to existing state-of-the-art rout-

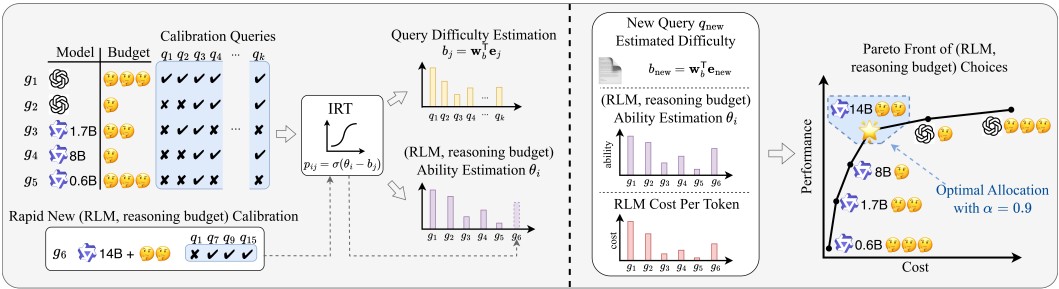

Figure 2: Illustration of our RADAR framework. **Left**: RADAR jointly estimates *interpretable* query difficulties and RLM configuration abilities using IRT (simplified for illustration purposes; full details in Section 3.3). New RLM configurations can be rapidly added by estimating their ability on a small subset of *dynamically selected* queries using adaptive testing (Section 3.5). **Right**: RADAR formulates routing as multi-objective optimization and routes queries to sufficiently capable configurations, optimizing performance-cost tradeoffs towards the Pareto front (Section 3.2).

ing methods. For example, on MATH-500 (Hendrycks et al., 2021c), RADAR can match $90\%$ of the performance of OpenAI o4-mini with a high reasoning budget at $1.31\%$ of its cost. RADAR exhibits strong generalization to OOD queries, including long-context multi-document QA (Krishna et al., 2024), despite being primarily trained on shorter queries. Further, RADAR scales and generalizes well to new RLM configurations, showing an improvement in routing performance. We summarize our key contributions below.

**[C1]** We cast adaptive reasoning as routing over discretized model–budget configurations and select configurations via a Pareto-optimal performance–cost objective, all in a black-box setting.

**[C2]** RADAR adapts item response theory to learn interpretable query difficulties and configuration abilities from data, enabling low-latency routing and generalization to unseen queries.

**[C3]** RADAR supports plug-and-play integration of new reasoning models via adaptive calibration that estimates abilities from a small, informative subset of queries.

**[C4]** Across 8 challenging reasoning benchmarks, RADAR achieves superior performance–cost tradeoffs and strong out-of-distribution generalization, including long-context document QA tasks.

## 2 RELATED WORK

**Efficient Reasoning.** A rapidly growing literature seeks to make reasoning models more efficient; see Yue et al. (2025) for an overview. Methods such as L1 (Aggarwal & Welleck, 2025) and S1 (Muennighoff et al., 2025) provide *length control*, encouraging shorter reasoning chain-of-thoughts that lead to correct answers. Others prune or adapt the reasoning process by dynamically shortening or extending reasoning (Hou et al., 2025; Xu et al., 2025; Wang et al., 2025); adaptively controlling inference steps (Huang et al., 2025); and analyzing when additional reasoning is beneficial or wasteful (Su & Cardie, 2025; Su et al., 2025; Yu et al., 2025; Ghosal et al., 2025). Our approach is complementary to single-model efficiency: RADAR can include these efficient RLMs as an additional model configuration routing candidates. In contrast to *static* single-model tuning, which requires access to RLM weights, RADAR works in a black-box setting, leveraging the complementary performance of multiple RLMs in a rapidly growing heterogeneous RLM landscape, and *dynamically* shifts performance-cost tradeoffs depending on user applications.

**Routing for Foundation Models.** Recent work in model-routing (Ding et al., 2024; Ong et al., 2024; Chen et al., 2025; Hu et al., 2024; Zhang et al., 2025) focuses on *model selection* with black-box predictors or model-cascades (Chen et al., 2023). In contrast, we explore *adaptive reasoning* through the lens of routing over *model–budget configurations* of RLMs and provide a novel formulation of routing as an MOO. Unlike opaque routing regressors (Chen et al., 2023; Ding et al., 2024; Ong et al., 2024), we employ an IRT parameterization to model latent query difficulties and model configuration abilities as *interpretable* parameters. Compared to a concurrent IRT-based routing method (Song et al., 2025), our work (1) provides a novel problem formulation of routing, on previously unexplored reasoning models, as an MOO opening a powerful toolkit of MOO solu-

tion techniques like Chebyshev scalarization, (2) uses a different IRT parameterization with fewer parameters providing an interpretable scalar-valued ability ordering among models and potentially requiring less training data, (3) provides new MOO-based routing performance metrics evaluating the coverage of the Pareto front, and (4) presents an adaptive-testing based method to quickly generalize the routing framework to new RLMs for improved performance. We provide an expanded related work section and detailed comparison in Appendix A.

## 3 METHODOLOGY

In this section, we introduce our RADAR framework for adaptive reasoning through routing RLM configurations. We begin by formulating adaptive reasoning as MOO, where the router selects the optimal {RLM, reasoning budget} configuration for a chosen performance-cost tradeoff. Under this formulation, we then detail (1) how to estimate a particular RLM's performance for a given query using IRT and (2) how to solve this optimization problem.

### 3.1 ROUTING-BASED ADAPTIVE REASONING IN RLMS THROUGH DISCRETIZATION TRICK

Unlike classic model routing (Ong et al., 2024; Ding et al., 2024), which chooses from a set of different base models, choosing the right RLM for practical deployment involves a performance and cost trade-off at two key levels: base models and reasoning budgets. We *unify* these decisions by *discretizing* each RLM $m \in \mathcal{M}$ by its available set of reasoning budgets $u \in \mathcal{U}_m$. For example, the available reasoning budgets can be {low, medium, high} for proprietary RLMs (e.g., OpenAI o4-mini), or a user-defined discrete set of values such as {0, 1k, 2k, 4k, 8k, 16k}, for open-source RLMs. To enforce a reasoning budget on an open-source RLM (e.g., Qwen3), we count the number of thinking tokens during generation. If this number exceeds the specified budget, we append an interruption message (e.g. "output answer based on current thinking") along with the RLM's end of thinking token (e.g., </think>) to complete the thinking chain-of-thought and preemptively start generating answer completion tokens. Each discretization is referred to as a model configuration $g = (m, u) \in \mathcal{G}$ with $\mathcal{G} \subseteq \bigcup_{m \in \mathcal{M}} \bigcup_{u \in \mathcal{U}_m} \{(m, u)\}$ being the set of all model configurations. Discretization helps enable routing in RLMs at the configuration level. We use the general term "configuration" here since, apart from the reasoning budget, RADAR can be used to select among other model settings, such as parameterizations of the RAG pipeline attached to the RLM or decoding methods employed.

### 3.2 FORMULATING RLM ROUTING AS A MULTI-OBJECTIVE OPTIMIZATION PROBLEM

We present a novel view of model routing through the lens of MOO, allowing us to leverage effective solution techniques from MOO literature (Miettinen, 1999; Branke et al., 2008; Murata & Ishibuchi, 1995; Zhang & Golovin, 2020). Given a set of queries $\mathcal{Q} = \{q_1, \ldots, q_k\}$ and a set of candidate model configurations $\mathcal{G} = \{g_1, \ldots, g_n\}$, our goal is to assign each query $q_j \in \mathcal{Q}$ to the optimal configuration $g_i \in \mathcal{G}$ which maximizes performance and minimizes cost. For each query $q$ (index $i$ dropped for brevity), we define an MOO with two objective functions: performance and cost. The performance prediction function $p_q : \mathcal{G} \to [0, 1]$ predicts the probability of a correct response by running configuration $g$ on query $q$. Similarly, the cost prediction function $c_q : \mathcal{G} \to [0, 1]$ predicts the cost of running configuration $g$ on query $q$, normalized to $[0, 1]$. We formulate the optimization problem as a two-dimensional a-priori MOO (Branke et al., 2008) and solve it using scalarization techniques (Murata & Ishibuchi, 1995; Zhang & Golovin, 2020), written as:

$$g^* = \arg\max_{g \in \mathcal{G}} \ f(p_q(g), c_q(g)), \tag{1}$$

where $f$ is a scalarization function. Scalarization aggregates the objective functions of an MOO to solve a single-objective problem (SOP), such that the optimal solutions to the SOP are Pareto optimal solutions to the MOO. We explore two scalarization techniques: linear scalarization (Murata & Ishibuchi, 1995) and Chebyshev scalarization (Zhang & Golovin, 2020).

**Linear Scalarization.** Linear scalarization (Murata & Ishibuchi, 1995) uses non-negative weights for each objective function of the MOO, and *maximizes* the weighted sum of *rewards* (or equivalently minimizes the weighted sum of costs). By using different weights in the aggregation, we can obtain different points on the performance-cost Pareto front. The linear scalarization problem (LSP) of our

MOO with weight vector $\boldsymbol{w} \in \mathbb{R}^2_{\geq 0}$ is given by:

$$\underset{g \in \mathcal{G}}{\arg\max} \ w_1 p_q(g) + w_2(-c_q(g)). \tag{2}$$

Since we can factor a multiplicative constant out of the weights, we use a weight vector that sums to 1. Further, since we have a two-dimensional MOO, we can simply set $w_2 = 1 - w_1$. Our LSP becomes:

$$\text{LSP}_q^{w_1} = \underset{g \in \mathcal{G}}{\arg\max} \ w_1 p_q(g) - (1 - w_1)c_q(g). \tag{3}$$

We note that Equation 3 recovers the routing formulation presented in existing routing methods (Hu et al., 2024; Song et al., 2025; Zhang et al., 2025) but arrived at through the lens of an MOO.

**Chebyshev Scalarization.** In general, if the Pareto front is non-convex, there could be points on the Pareto front that cannot be obtained as the solutions of any weight-parameterized LSP. We therefore also explore Chebyshev scalarization (Zhang & Golovin, 2020), which uncovers points in the concave parts of the Pareto front by formulating the SOP as a weighted Chebyshev distance to an ideal reference point. In our case, the ideal reference point has a performance of 1 and a cost of 0. Chebyshev scalarization aims to *minimize* the maximum weight-scaled *penalty* over all dimensions of the MOO from an ideal reference point. The Chebyshev scalarization problem (CSP) of our MOO with weight vector $\boldsymbol{w} \in \mathbb{R}^2_{\geq 0}$ is given by:

$$\text{CSP}_q^{w_1} = \underset{g \in \mathcal{G}}{\arg\min} \max\{w_1|1 - p_q(g)|, \ (1 - w_1)c_q(g)\}. \tag{4}$$

The weight parameter $w_1$ controls the trade-off between performance and cost: a larger value of $w_1$ means a preference for performance over cost by favoring stronger model configurations more often, while a smaller value of $w_1$ prefers weaker but more cost-effective model configurations. Given a user-specified tradeoff profile with weight $w_1$ and query $q$, RADAR assigns configuration $g = \text{LSP}_q^{w_1}$ (or configuration $g = \text{CSP}_q^{w_1}$ depending on the chosen scalarization scheme) to maximize performance and minimize cost.

### 3.3 IRT-BASED CALIBRATION OF RLM REASONING ABILITY AND QUERY DIFFICULTY

A key component in solving the MOO in Equation 1 is an accurate parameterization of the performance prediction function $p_q(g)$, which predicts the probability of a correct response by configuration $g$ on query $q$. We leverage IRT (Rasch, 1960; Lord, 2012; van der Linden & Hambleton, 1997; DeMars, 2010) that is often used to model student responses to test items, specifically the two-parameter logistic (2PL) model (Lord, 1951; Birnbaum, 1968), to parameterize our performance prediction function. IRT assumes monotonicity, i.e., as a model configuration's ability increases, its probability of correctly answering a query also increases. The 2PL model takes two query characteristics into account, *difficulty* and *discrimination*. Intuitively, a configuration's *ability* estimate is impacted differently after it answers a query correctly, depending on the difficulty of the query. Query discrimination encodes the varying rate at which the likelihood of a correct response increases with the model configuration's ability.

In RADAR, we embed query $q_j$ into a $d_q$-dimensional vector $\boldsymbol{e}_j$ using a frozen embedding model. Leveraging the content of the queries through embeddings helps RADAR generalize to OOD queries, including ones from long-context multi-document QA, despite being trained on shorter queries. We obtain the scalar-valued difficulty $b_j \in \mathbb{R}$ and discrimination $a_j \in \mathbb{R}$ by linear transformations of the query embeddings, i.e., $a_j = \boldsymbol{w}_a^\mathsf{T} \boldsymbol{e}_j, b_j = \boldsymbol{w}_b^\mathsf{T} \boldsymbol{e}_j$, where $\boldsymbol{w}_a, \boldsymbol{w}_b \in \mathbb{R}^{d_q}$ are learnable $d_q$-dimensional transformation vectors. Our design choice of simple linear transformations and a frozen embedding model helps ensure minimal router latency, which we analyze in Section 4.3 and Appendix B.1. We use a scalar-valued ability parameter $\theta_i \in \mathbb{R}$ for model configuration $g_i$. In the 2PL model, the probability that a model configuration $g_i$ correctly answers a query $q_j$ is given by $p_{ij} = \sigma(a_j(\theta_i - b_j))$ where $\sigma(\cdot)$ is the sigmoid function.

We note that a concurrent work in IRT-based model-routing (Song et al., 2025) uses a multi-dimensional IRT model (MIRT) (Reckase, 2009) to parameterize the performance function. In contrast, we use scalar-valued model configuration abilities, enabling learned ability values to capture ordering information among model configurations and thus be interpretable. The fewer number of

parameters in the 2PL model means that it requires less data to train than an MIRT model. Further, instead of the qualitative model profile embedding approach to model generalization adopted by Song et al. (2025), we use an adaptive testing-based approach to quickly estimate the precise scalar ability of any new RLM configuration, enabling RADAR to rapidly include it in the pool of RLM configurations for routing for future queries (see Section 3.5).

To train the IRT model, we construct a binary-valued evaluation matrix $Y \in \{0, 1\}^{n \times k}$ of responses by $n$ RLM configurations, i.e., different models with different reasoning budgets, on a set of $k$ training queries. We minimize the average negative log-likelihood (binary cross-entropy) over all $nk$ observed responses, given by:

$$\mathcal{L}_{\text{2PL}} = -\frac{1}{nk} \sum_{i=1}^{n} \sum_{j=1}^{k} \big[ y_{ij} \log p_{ij} + (1 - y_{ij}) \log(1 - p_{ij}) \big], \tag{5}$$

where $p_{ij} = \sigma(a_j(\theta_i - b_j))$ denotes the probability under the 2PL IRT model that configuration $g_i$ correctly answers a query $q_j$.

## 3.4 COST PREDICTION

Apart from the performance prediction function, the other component in solving our routing MOO (see Equation 1) is the cost prediction function $c_q(g)$. Following prior routing work (Hu et al., 2024; Song et al., 2025), we adopt a heuristic-based approach. We calculate the output cost of using configuration $g$ to answer query $q$ by multiplying the cost per token $t_g$ of the base RLM of configuration $g$, with the total number of reasoning tokens $n_{g(q)}^{\text{rsn}}$ and completion tokens $n_{g(q)}^{\text{cmp}}$ generated, which is then averaged over all training queries. This heuristic is given by:

$$c_q(g) = 1/|\mathcal{Q}| \sum_{q \in \mathcal{Q}} (n_{g(q)}^{\text{rsn}} + n_{g(q)}^{\text{cmp}}) \, t_g, \tag{6}$$

where we obtain the cost per token $t_g$ in US dollars of the base RLM from the official website for proprietary models (e.g., OpenAI for o4-mini) or from cloud providers[1] for open source models (e.g., Qwen3). We rescale (using min-max normalization) each configuration cost $c_q(g)$ to the range $[0, 1]$, to ensure that both predicted cost and performance are on the same scale. We leave the development of more elaborate cost prediction methods to future work.

## 3.5 EXPANDING THE POOL OF RLM CONFIGURATIONS THROUGH ADAPTIVE TESTING

To add a new model configuration $g_i$ to RADAR, we need an accurate estimate of its ability $\hat{\theta}_i \in \mathbb{R}$. Given a fitted 2PL model with learned query parameters $\{a_j, b_j\}_{j=1}^{|\mathcal{Q}|}$, we could estimate the ability $\hat{\theta}_i$ of $g_i$ by observing its responses to a query subset $\mathcal{S} \subseteq \mathcal{Q}$. We define the corresponding negative log-likelihood as:

$$\mathcal{L}_{\text{2PL}}(\theta; \mathcal{S}, g_i) = -\sum_{j \in \mathcal{S}} \big[ y_{ij} \log \sigma(a_j(\theta - b_j)) + (1 - y_{ij}) \log(1 - \sigma(a_j(\theta - b_j))) \big], \tag{7}$$

where $y_{ij} \in \{0, 1\}$ denotes the response of configuration $g_i$ to query $q_j$. The ability estimate is obtained by minimizing the negative log-likelihood, given by $\hat{\theta}_i = \arg\min_\theta \mathcal{L}_{\text{2PL}}(\theta; \mathcal{S}, g_i)$.

A complete evaluation corresponds to $\mathcal{S} = \mathcal{Q}$, yeilding $\hat{\theta}_i = \arg\min_\theta \mathcal{L}_{\text{2PL}}(\theta; \mathcal{Q}, g_i)$. However, evaluating on all queries is computationally expensive. To reduce evaluation cost, we adopt an adaptive query selection strategy inspired by computerized adaptive testing (Wainer et al., 2000). We iteratively construct an evaluation subset $\mathcal{S}_t$ for configuration $g_i$. For clarity, we suppress the configuration index $i$ during the iterative procedure. We initialize $\mathcal{S}_0 = \varnothing$ and $\hat{\theta}_0 = 0$. For iteration $t \geq 1$, inspired by Hofmann et al. (2025), we select the query with the highest Fisher information on the current ability estimate, given by:

$$j_t = \underset{j \in \mathcal{Q} \setminus \mathcal{S}_{t-1}}{\arg\max} I(\hat{\theta}_{t-1}, a_j, b_j), \tag{8}$$

where the Fisher information under the 2PL IRT model is given by:

$$I(\theta, a_j, b_j) = a_j^2 \sigma(a_j(\theta - b_j))[1 - \sigma(a_j(\theta - b_j))]. \tag{9}$$

---

[1] https://www.together.ai/pricing

Table 1: Routing performance on ID queries across benchmarks reported on the hypervolume metric (higher is better). RADAR outperforms baselines, denoting better performance-cost tradeoffs towards the Pareto front. See Table 13 for performance on OOD queries. Best performance is in **bold** and second best is underlined.

| Benchmark | Random-Pair | RouterBench | IRT-Router | RADAR (ours) |
|---|---|---|---|---|
| GPQA-Diamond | 0.5545 | 0.6866 | 0.6942 | **0.7513** |
| MMLU | 0.6905 | 0.8592 | 0.8604 | **0.8720** |
| MMLU-Redux | 0.7281 | 0.9053 | 0.9117 | **0.9230** |
| MMLU-Pro | 0.5589 | 0.7819 | 0.7812 | **0.7995** |
| LSAT | 0.6913 | 0.9125 | 0.9163 | **0.9188** |
| AIME | 0.5159 | 0.7680 | **0.7766** | 0.7760 |
| MATH-500 | 0.7433 | **0.9528** | 0.9420 | 0.9449 |
| FRAMES | 0.6589 | 0.8325 | 0.8501 | **0.8762** |

We update our evaluation subset with the selected query as $\mathcal{S}_t = \mathcal{S}_{t-1} \cup \{j_t\}$. After observing the response of configuration $g_i$ on the selected query $j_t$, we update its ability estimate as $\hat{\theta}_t = \arg\min_\theta \mathcal{L}_{2\mathrm{PL}}(\theta; \mathcal{S}_t, g_i)$. We repeat this procedure until the size of our evaluation subset $|\mathcal{S}_t|$ reaches a budgeted size $T$, returning $\hat{\theta}_T$ as the estimated ability of the new configuration $g_i$.

## 4 EXPERIMENTAL EVALUATION

In this section, we detail our evaluation setup, including benchmarks, metrics, and baselines, used for a comprehensive evaluation of RADAR.

**Benchmarks.** We evaluate on eight reasoning benchmarks, including **AIME** (MAA, 2024), **MATH** (Hendrycks et al., 2021c), **GPQA** (Rein et al., 2024), **LSAT** (Wang et al., 2022; Zhong et al., 2021), **MMLU** (Hendrycks et al., 2021b;a), **MMLU Redux** (Gema et al., 2024), **MMLU Pro** (Wang et al., 2024), and **FRAMES** (Krishna et al., 2024), spanning competition math, PhD-level science, law, and general knowledge with both multiple-choice and open-ended formats. In particular, **FRAMES** probes RADAR's generalization to long-context queries in a multi-doc QA setting; long-context evaluation is largely absent in prior routing work-RouterBench (Hu et al., 2024) is a notable exception, but on a private in-distribution RAG set. See Appendix C for details.

**Metrics.** We report **hypervolume** (Emmerich & Deutz, 2018), which, in our setting, corresponds to the area under the performance-cost trade-off curve across weights $w_1$ (higher is better). We also use the cost–performance threshold (**CPT**) metric, akin to call-performance thresholds (Ong et al., 2024): relative to the best-performing configuration, o4-mini–high, $\mathrm{CPT}(x\%)$ is the minimum cost required to reach $x\%$ of that performance, normalized by the cost of o4-mini–high. For example, $\mathrm{CPT}(90\%) = 0.1$ means achieving 90% performance at 10% of the cost.

**Baselines.** We compare RADAR to several recent, state-of-the-art model routing methods, including **RouterBench** (Hu et al., 2024; Chen et al., 2025) and **IRT-Router** (Song et al., 2025). These methods do not natively generalize for adaptive reasoning or RLM configuration routing. We therefore adapt these methods for our experimental setting; details are available in Appendix D.3. We also compare to several heuristic-based baselines, including **All-Large** (o4-mini, high budget), **All-Small** (Qwen3 0.6B, 0 tokens), **Oracle** (chooses the cheapest best-performing configuration given test-set performance), and **Random-All** (uniform over configurations). **Random-Pair** selects the largest configuration with probability $w_1$, the user-defined performance–cost weight.

**Evaluation Setup and Implementation Details.** We employ Chebyshev scalarization in RADAR. We conduct both in-distribution (ID) and out-of-distribution (OOD) evaluations. For ID experiments, we aggregate the training splits of all 8 benchmarks into a single training set for training the 2PL IRT model (see Section 3.3) and report performance on the test split of each benchmark separately. We use an $80\% - 20\%$ train-test split for benchmarks without a predefined test set. For OOD, for each benchmark, we aggregate the training splits of the other remaining *non-overlapping* benchmarks into a single training set and report performance on the test split of this benchmark. For example, for the OOD experiment on AIME, the training split of AIME and of overlapping benchmarks (MATH since MATH includes questions from AIME), are held out from the training set.

Table 2: Routing performance on ID queries across benchmarks reported on the CPT $(90\%)$ metric (lower is better). CPT $(90\%)$ denotes the fraction of the cost of running OpenAI o4-mini with a high reasoning budget to match $90\%$ of its performance. See Table 14 for performance on OOD queries. Best performance is in **bold** and second best is underlined.

| Benchmark | Random-Pair | RouterBench | IRT-Router | RADAR (ours) |
|---|---|---|---|---|
| GPQA-Diamond | 80.36% | 57.13% | 53.99% | **13.21%** |
| MMLU | 76.30% | 2.71% | **2.66%** | 2.69% |
| MMLU-Redux | 75.06% | 2.59% | 2.80% | **2.42%** |
| MMLU-Pro | 83.57% | 5.19% | **3.83%** | 3.89% |
| LSAT | 80.14% | 2.02% | 1.93% | **1.82%** |
| AIME | 87.22% | 65.65% | 61.23% | **60.69%** |
| MATH-500 | 76.15% | 1.34% | 1.44% | **1.31%** |
| FRAMES | 77.90% | 43.50% | 31.53% | **13.11%** |

Table 3: Routing performance across benchmarks reported on the hypervolume metric (higher is better), before (RADAR) and after (RADAR++) adding new RLM configurations from Qwen3-14B, to test RADAR's model generalization capability. RADAR++ quickly estimates the abilities of new configurations through adaptive testing for an improved routing performance. Best performance is in **bold** and second best is underlined.

| Benchmark | In-Distribution (ID) | | Out-of-Distribution (OOD) | |
|---|---|---|---|---|
| | RADAR | RADAR++ | RADAR | RADAR++ |
| GPQA-Diamond | 0.7513 | 0.7535 | 0.7466 | 0.7463 |
| MMLU | 0.8720 | 0.8731 | 0.8609 | 0.8698 |
| MMLU-Redux | 0.9230 | 0.9238 | 0.9072 | 0.9091 |
| MMLU-Pro | 0.7995 | 0.8021 | 0.7858 | 0.7951 |
| LSAT | 0.9188 | 0.9233 | 0.9146 | 0.9255 |
| AIME | 0.7760 | 0.7828 | 0.7566 | 0.7566 |
| MATH-500 | 0.9449 | 0.9461 | 0.9368 | 0.9368 |
| FRAMES | 0.8762 | 0.8830 | 0.8865 | 0.8931 |

We route over 35 configurations comprising OpenAI **o4-mini** (budgets: low, medium, high) and **Qwen3** models (0.6B/1.7B/4B/8B) with budgets 0, 256, 512, 1k, 2k, 4k, 8k, 16k (Yang et al., 2025; OpenAI, 2025). Each configuration is evaluated once per training query with standard prompts (Appendix D.4); for AIME's small test set, we average over eight runs. Results closely match reported RLM performance (Yang et al., 2025; OpenAI, 2025). In total, we collected 1.75 million binary responses over 50,139 unique questions across train/test splits of all eight benchmarks. Further details are in Appendix D.

## 4.1 MAIN QUANTITATIVE RESULTS

**RADAR outperforms state-of-the-art model routing methods.** Table 1 and Table 2 report routing performance of all methods across 8 reasoning benchmarks evaluated in the ID setting on the hypervolume and CPT(90%) metrics, respectively. We include results on the CPT metric at additional thresholds in Appendix B.2. We see that RADAR outperforms all baselines on most benchmarks and performs comparably to the best existing baseline on the remaining benchmarks. RADAR outperforms IRT-Router (Song et al., 2025), a concurrent IRT-based routing work, suggesting that our novel formulation of RLM routing as an MOO, as well as the use of solution techniques like Chebyshev scalarization, enable better recovery of the Pareto performance-cost front. For example, on the challenging GPQA-Diamond benchmark, RADAR demonstrates an $8\%$ performance boost over the second-best baseline on the hypervolume metric. On the CPT metric, on MATH-500, RADAR is able to match $90\%$ of the performance of o4-mini with a high reasoning budget at $1.31\%$ of its cost. Similar gains are seen across all benchmark tasks.

**RADAR exhibits strong query generalization capabilities.** We report routing performance of all methods across 8 reasoning benchmarks evaluated in the OOD setting on the hypervolume and CPT $(90\%)$ metrics, respectively, in Appendix B.3. We show the Pareto performance-cost tradeoff curves for all methods on OOD queries from FRAMES (Krishna et al., 2024) in Figure 1. We see that RADAR exhibits strong generalization to OOD queries, outperforming existing state-of-the-art methods on most benchmarks. In particular, we highlight its dominant performance on the challenging long-context, multi-document reasoning-based QA task from FRAMES (Krishna et al.,

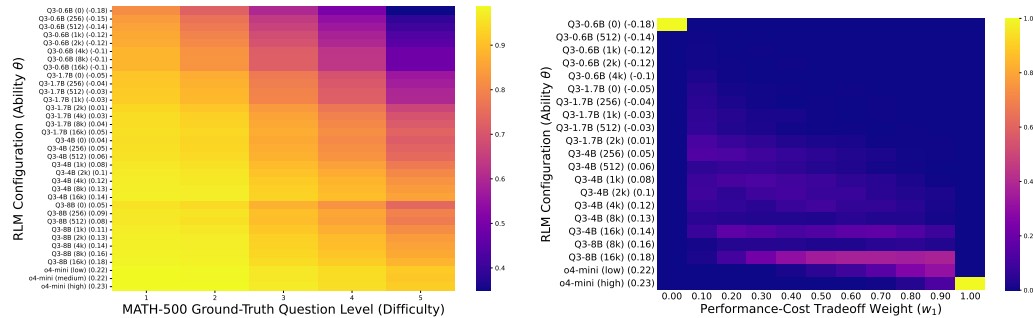

Figure 3: RADAR estimates *interpretable* query difficulties and RLM configuration abilities. **Left:** Mean predicted correctness probability of configurations on questions with 5 different ground-truth difficulty levels in MATH-500. As difficulties increase, configurations with higher abilities are predicted to perform better. **Right:** Fraction of routing calls on MATH-500 queries spread across RLM configurations when varying the performance-cost tradeoff weight. A lower (higher) weight leverages a higher fraction of Qwen3 (o4-mini) configurations, prioritizing cost (performance).

2024), despite primarily being trained on much shorter queries. When generalizing to OOD queries with significantly higher difficulty (e.g., AIME) than those seen during training, RADAR tends to assign a model configuration with a slightly lower ability than optimal, resulting in a slight decrease in performance. This weakness can be addressed by including a small number of representative queries during training, as shown in Appendix B.4.

**Ablation study.** In Appendix B.5, we show that Chebyshev scalarization outperforms linear scalarization, which is adopted in prior routing work (Song et al., 2025; Hu et al., 2024), in the OOD experimental setting due to its ability to explore both convex and concave points on the Pareto front. In the ID experimental setting, both scalarization techniques perform similarly, with linear being marginally better. Our novel formulation of routing as an MOO opens the door to leveraging other MOO solution techniques in future work. We also conduct an ablation on the size of the training matrix. Using just 20% of subsampled training queries, RADAR achieves a similar performance to using the entire training set as shown in Appendix B.6.

## 4.2 MODEL SCALABILITY AND GENERALIZATION EVALUATION

We evaluate the scalability of RADAR to new RLMs by adding 8 new model configurations from the Qwen3 14B RLM. Table 3 shows the result; using adaptive testing, RADAR accurately estimates the abilities of these new configurations by dynamically selecting just 5k training queries ( 12% of the training set) for evaluation, resulting in improved routing performance. We compare against uniform random sampling in Appendix B.7. We show how routing shifts to new RLM configurations in Appendix B.8. In contrast to a concurrent IRT-based routing method (Song et al., 2025), which embeds a qualitative profile of the new model by prompting ChatGPT, or even existing model-pair-based routing methods (Ding et al., 2024), which assume the new model-pair has a similar ability difference, RADAR can accurately estimate the ability of any new model configuration.

## 4.3 INTERPRETABILITY AND LATENCY ANALYSIS

**RADAR estimates interpretable query difficulties and RLM configuration abilities.** We highlight the interpretable nature of RADAR through a case study on ID queries from MATH-500 (Hendrycks et al., 2021c) where queries are annotated with one of five levels of increasing difficulty. On the left panel of Figure 3, we find a moderate Pearson correlation coefficient of 0.509 between RADAR-estimated query difficulties and the five ground-truth levels. With an increase in query level, we see that configurations with higher abilities are predicted to have higher correctness probabilities, leading to a mean answer correctness prediction accuracy of 84.42% over all configurations and queries. On the right panel of Figure 3, we show the fraction of routing calls across configurations as the performance-cost tradeoff weight is varied, with cost-effective Qwen3 models preferred at lower weights and performant o4-mini models preferred at higher weights.

**RADAR works in real time with minimal latency overhead.** We measure the latency of RADAR and compare it to the latency of the smallest RLM configuration (Qwen3-0.6B with 0 reasoning

budget) used to generate answers to queries. The average per query routing latency overhead of RADAR over three runs of $500$ queries from MATH-500 (Hendrycks et al., 2021c) is $6.89 \pm 0.53$ milliseconds. Compared to the time taken for the smallest RLM configuration to answer the query, which is $869.56 \pm 1.1$ milliseconds, RADAR adds negligible overhead. We analyze throughput in Appendix B.9.

## 5 Conclusions and Future Work

We introduced RADAR, a reasoning–ability and difficulty-aware routing framework that (1) formalizes adaptive reasoning as an MOO and (2) leverages item response theory to adaptively assign queries to RLM model–budget configurations. RADAR achieves strong cost–performance tradeoffs, consistently outperforming prior routing methods across eight challenging reasoning benchmarks, and generalizes well to out-of-distribution queries. Beyond efficiency, RADAR offers interpretability by exposing query difficulty and model abilities, and supports plug-and-play integration of new RLM configurations through adaptive calibration. Several promising avenues for future work exist. First, we would like to extend RADAR beyond text to multi-modal reasoning settings. Second, incorporating additional configurations beyond the reasoning budget, such as retrieval, tool usage, and decoding algorithms, may yield fine-grained routing decisions for a wider range of applications, such as ultra-long context QA and deep research. Third, exploring RADAR in other constraint scenarios, such as when there is a total budget constraint on a batch of queries. Together, these directions highlight the broader potential of RADAR as a principled, interpretable foundation for adaptive reasoning in an ever-evolving RLM ecosystem.

## Acknowledgements

The authors would like to thank Tong Sun and Jaewook Lee for helpful discussions. We also thank the anonymous reviewers for their helpful comments.

## Reproducibility Statement

We structured the paper and the Appendix so the results can be independently re-implemented and reproduced. The problem setup, routing objective, and IRT modeling details are specified in Section 3. Dataset sources, preprocessing, and ID/OOD split procedures are detailed in Appendix C. Experimental details such as baselines, evaluation protocol, metrics, and other implementation details are available in Section 4 and Appendix D.

## Ethics Statement

We affirm adherence to the ICLR Code of Ethics. Our study evaluates routing over public reasoning benchmarks and does not involve human subjects or personally identifiable data; we follow dataset licensing and attribution guidance and document detailed preprocessing steps in Appendix C. We note that any routed answer inherits the properties, including potential social biases or safety issues, of the underlying RLM configurations; our method does not itself mitigate these risks, and we caution against deployment without domain-appropriate safeguards and bias auditing.

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

## A  EXTENDED RELATED WORK

**Efficient Reasoning.**  A rapidly growing literature seeks to make reasoning models themselves more efficient; see (Yue et al., 2025) for a broader overview of this direction. Methods such as L1 (Aggarwal & Welleck, 2025) and S1 (Muennighoff et al., 2025) provide *length control*, enabling reasoning models to trade off accuracy and cost by constraining chain-of-thought length. Others prune or adapt the reasoning process by dynamically shortening or extending reasoning (Hou et al., 2025; Xu et al., 2025; Wang et al., 2025); adaptively controlling inference steps (Huang et al., 2025); and analyzing when additional reasoning is beneficial or wasteful (Su & Cardie, 2025; Su et al., 2025; Yu et al., 2025; Ghosal et al., 2025). Theoretical perspectives further study optimal reasoning length (Lee et al., 2025). These works aim to make a single model more efficient. They also require access to model weights, which usually do not apply for closed-source or black-box settings. Our approach is complementary: RADAR treats any such efficient reasoning model as an additional candidate in its pool of (model, reasoning budget) configurations. This means advances in adaptive or efficient reasoning can be seamlessly integrated into our framework, while RADAR contributes orthogonally by providing per-query routing, interpretability through IRT, and Pareto-optimal cost–performance control. In contrast to *static* single-model tuning, which requires weight access, RADAR operates in a black-box setting and leverages the complementary strengths of diverse RLMs. This enables RADAR to *dynamically* shift along performance–cost tradeoffs depending on application needs in a heterogeneous RLM landscape.

**Routing for Foundation Models.**  Recent work studies cost–quality routing across multiple LLMs (Chen et al., 2023; Zhang et al., 2024; Ding et al., 2024; Ong et al., 2024; Hu et al., 2024; Šakota et al., 2024; Chen et al., 2025; Song et al., 2025). Most methods focus on *model selection* with black-box predictors or cascades (Chen et al., 2023; Ding et al., 2024; Ong et al., 2024; Šakota et al., 2024; Chen et al., 2025), though TREACLE additionally co-selects prompt types under budget constraints (Zhang et al., 2024). We instead study *adaptive reasoning* and cast this problem as routing over *model–budget configurations*, where the budget controls the number of thinking tokens, making reasoning cost an explicit decision dimension in addition to the model itself. Routers also differ in *when* they commit: cascaded approaches may re-query a model (Chen et al., 2023; Zhang et al., 2024), while others choose once per query (Ding et al., 2024). Our router makes a single assignment before generation, avoiding mid-turn switching (and KV-cache recomputation) or multiple re-querying while still retaining favorable cost–quality trade-offs. Finally, we emphasize *interpretability and control*. Unlike opaque regressors (Chen et al., 2023; Ding et al., 2024; Ong et al., 2024), we use an IRT parameterization to expose query difficulty and configuration ability.

**Comparison with IRT-Router**  Compared to IRT-Router (Song et al., 2025), a concurrent and recently released work, RADAR makes the following contributions:

1. Novel MOO Formulation: We're the first to formulate model routing in a mathematically principled way as multi-objective optimization (MOO) that searches for the model at the Pareto front of the performance-cost tradeoff curve using scalarization techniques. The objective used in IRT-Router is introduced ad hoc and is a *special case* of our MOO formulation with linear scalarization. Beyond simple linear scalarization, which cannot recover non-convex Pareto fronts, our MOO formulation allows the LLM routing community to leverage powerful solution techniques from well-established MOO literature, including Chebyshev scalarization. In RADAR, we find that Chebyshev scalarization outperforms linear scalarization on the stable hypervolume metric, in the challenging OOD experimental setting, due to its ability to explore both convex and concave points on the Pareto front (see Table 18). In the easier ID experimental setting, both scalarization techniques perform similarly, with linear being marginally better. We leave the exploration of other MOO solution techniques, such as lexicographic methods, for future work.

2. Model Generalizability: Another significant contribution of RADAR is its effective model generalization capability. IRT-Router uses an ad hoc approach: it queries ChatGPT (web search mode) for a description of the new LLM, which is then corrected by *manual intervention* and embedded. IRT-Router notes the limited generalizability of their approach and highlights model generalizability as an important direction for future work. In contrast, to add a new RLM configuration in RADAR, we simply need its ability. To precisely estimate

Table 4: Dataset statistics of prompt tokens across reasoning benchmarks used.

| Dataset | Samples | Mean Tokens | Min Tokens | Max Tokens |
|---|---|---|---|---|
| AIME | 1,035 | 143.00 | 30 | 3,312 |
| MATH | 8,000 | 86.16 | 24 | 806 |
| GPQA | 448 | 250.27 | 82 | 2,812 |
| LSAT | 2,025 | 263.63 | 174 | 570 |
| MMLU | 13,937 | 150.50 | 66 | 1,040 |
| MMLU Redux | 5,298 | 135.50 | 68 | 1,000 |
| MMLU Pro | 12,032 | 237.51 | 70 | 1,700 |
| FRAMES | 561 | 16,272.19 | 690 | 31,954 |

the ability of a new RLM configuration, RADAR follows a *principled and fully automated* approach by evaluating it on a small set (12% of training queries) of dynamically selected queries, employing a classic technique inspired by adaptive testing in educational assessment.

3. Informative Metrics: IRT-Router reports performance at three arbitrary performance-cost tradeoff weights (0.2, 0.5, and 0.8), which fail to provide an accurate evaluation of routing methods. In contrast, our MOO-based routing formulation leverages the hypervolume metric from MOO literature, which corresponds to the area under the performance-cost trade-off curve across the entire domain (0 to 1) of the trade-off weight, yielding a more stable and informative metric.

4. Custom Interpretable IRT Model: IRT-Router employs a standard Multidimensional-IRT (MIRT) model, which uses non-interpretable vectorized abilities. In contrast, RADAR uses a custom, interpretable IRT model with multidimensional embeddings for queries to enable OOD generalizability and scalar model abilities to support *interpretable* ability ordering across models. This ability ordering, as seen in the y-axis of Figure 3, is helpful for RLM benchmarking and evaluation.

5. Focus on Reasoning LLMs: In contrast to past work, including IRT-Router, which are primarily focused on LLMs, we present a routing formulation for RLMs incorporating reasoning budgets. Choosing the right RLM for practical deployment involves a performance-cost trade-off at two key levels: base models and reasoning budgets. We *unify* these decisions by *discretizing* each RLM by its available set of reasoning budgets. Although simple, our discretization trick easily extends to selecting other model settings, such as parameterizations of the RAG pipeline attached to the RLM or decoding methods employed.

**Item Response Theory in Machine Learning.** Originally designed for assessment and other educational applications, IRT has emerged as a versatile tool for understanding and improving foundation models. It has been applied to evaluation and benchmarking such as jointly estimating model ability and item difficulty to build adaptive or efficient test suites (Rodriguez et al., 2021; Zouhar et al., 2025; Hofmann et al., 2025; Polo et al., 2024); to training and curriculum design, where IRT-based difficulty estimates guide data selection for faster and more effective learning (Meng et al., 2024; Scarlatos et al., 2025); and to the diagnostics and bias analysis, exposing strengths and weaknesses of models relative to humans or ideological leanings (Gor et al., 2024). Most relevant to our setting, IRT has recently been explored for multi-model routing, where it parameterizes query difficulty and model ability to guide cost–performance trade-offs with interpretability (Song et al., 2025). Our work extends this line by applying IRT not only to model selection but also to adaptive reasoning configurations (model-reasoning budget pairs), thereby contributing to the ongoing exploration of IRT for foundation models.

# B  ADDITIONAL RESULTS

## B.1  ABLATION STUDY ON LINEAR TRANSFORM FOR QUERY DIFFICULTY

To test the sufficiency of our linear transformation for query difficulty and discrimination, we perform an ablation study with a classic two-layer multilayer perceptron (MLP) model using a ReLU

Table 5: Ablation study on the linear transformation for query difficulty in RADAR. Linear transformation performs similarly to a classic two-layer multilayer perceptron (MLP) model using a ReLU non-linearity on ID queries.

| Benchmark | Hypervolume (higher is better) | | CPT(90%) (lower is better) | |
|---|---|---|---|---|
| | RADAR | RADAR (MLP) | RADAR | RADAR (MLP) |
| GPQA-Diamond | 0.7513 | 0.7308 | 13.21% | 14.11% |
| MMLU | 0.8720 | 0.8707 | 2.69% | 2.8% |
| MMLU-Redux | 0.9230 | 0.9239 | 2.42% | 2.51% |
| MMLU-Pro | 0.7995 | 0.7955 | 3.89% | 3.91% |
| LSAT | 0.9188 | 0.9132 | 1.82% | 2.32% |
| AIME | 0.7760 | 0.7687 | 60.69% | 53.28% |
| MATH-500 | 0.9449 | 0.9365 | 1.31% | 1.33% |
| FRAMES | 0.8762 | 0.8777 | 13.11% | 11.37% |

Table 6: Ablation study on the linear transformation for query difficulty in RADAR. Linear transformation performs similarly to a classic two-layer multilayer perceptron (MLP) model using a ReLU non-linearity on OOD queries.

| Benchmark | Hypervolume (higher is better) | | CPT(90%) (lower is better) | |
|---|---|---|---|---|
| | RADAR | RADAR (MLP) | RADAR | RADAR (MLP) |
| GPQA-Diamond | 0.7466 | 0.7245 | 17.6% | 23.49% |
| MMLU | 0.8609 | 0.8573 | 2.63% | 2.82% |
| MMLU-Redux | 0.9072 | 0.9067 | 2.54% | 2.63% |
| MMLU-Pro | 0.7858 | 0.7844 | 3.54% | 4.82% |
| LSAT | 0.9146 | 0.9101 | 2.15% | 2.12% |
| AIME | 0.7566 | 0.7726 | 55.30% | 59% |
| MATH-500 | 0.9368 | 0.9424 | 1.55% | 1.46% |
| FRAMES | 0.8865 | 0.8771 | 9.99% | 11.17% |

non-linearity to obtain the difficulty and discrimination of queries. Our results reported in Table 5 and Table 6 show similar performance in both the ID and OOD experimental settings across benchmarks. These none to marginal gains, obtained at the cost of diminished interpretability and increased latency, further justify the simplicity of our design choices in RADAR.

## B.2 RESULTS ON THE CPT METRIC AT VARIOUS THRESHOLDS

In addition to $90\%$, we report results on the CPT metric at various thresholds, including $80\%$ (Table 7 and Table 8), $85\%$ (Table 9 and Table 10), and $95\%$ (Table 11 and Table 12), on both ID and OOD experimental settings across benchmarks. We observe similar patterns: RADAR outperforms baselines on a majority of datasets in both ID and OOD experimental settings.

Table 7: Routing performance on ID queries across benchmarks reported on the CPT $(80\%)$ metric (lower is better). CPT $(80\%)$ denotes the fraction of the cost of running OpenAI o4-mini with a high reasoning budget to match $80\%$ of its performance. Best performance is in **bold** and second best is underlined.

| Benchmark | Random-Pair | RouterBench | IRT-Router | RADAR (ours) |
|---|---|---|---|---|
| GPQA-Diamond | 62.69% | 17.69% | 8.4% | **5.05**% |
| MMLU | 52.61% | **1.22**% | 1.31% | 1.41% |
| MMLU-Redux | 52.93% | **1.32**% | 1.32% | 1.4% |
| MMLU-Pro | 68.24% | 1.7% | 1.74% | **1.37**% |
| LSAT | 60.9% | 1.47% | 1.36% | **1.08**% |
| AIME | 74.45% | 24.73% | 24.34% | **21.38**% |
| MATH-500 | 53.79% | **0.7**% | **0.7**% | 0.79% |
| FRAMES | 60.92% | 1.1% | 1.08% | **0.78**% |

Table 8: Routing performance on OOD queries across benchmarks reported on the CPT (80%) metric (lower is better). CPT (80%) denotes the fraction of the cost of running OpenAI o4-mini with a high reasoning budget to match 80% of its performance. Best performance is in **bold** and second best is underlined.

| Benchmark | Random-Pair | RouterBench | IRT-Router | RADAR (ours) |
|---|---|---|---|---|
| GPQA-Diamond | 66.05% | **5.2**% | 8.49% | 5.55% |
| MMLU | 52.58% | 1.45% | **1.23**% | 1.42% |
| MMLU-Redux | 52.5% | 1.47% | **1.29**% | 1.53% |
| MMLU-Pro | 66.14% | 1.73% | 1.82% | **1.4**% |
| LSAT | 61.0% | 1.44% | 1.36% | **1.13**% |
| AIME | 71.82% | — | **13.37**% | 30.5% |
| MATH-500 | 53.58% | 0.68% | 0.71% | **0.63**% |
| FRAMES | 60.73% | 1.33% | **1.05**% | 1.4% |

Table 9: Routing performance on ID queries across benchmarks reported on the CPT (85%) metric (lower is better). CPT (85%) denotes the fraction of the cost of running OpenAI o4-mini with a high reasoning budget to match 85% of its performance. Best performance is in **bold** and second best is underlined.

| Benchmark | Random-Pair | RouterBench | IRT-Router | RADAR (ours) |
|---|---|---|---|---|
| GPQA-Diamond | 71.53% | 37.41% | 31.19% | **8.93**% |
| MMLU | 64.46% | **1.61**% | 1.85% | 1.68% |
| MMLU-Redux | 63.77% | 1.85% | 1.97% | **1.67**% |
| MMLU-Pro | 75.91% | 2.18% | **2.14**% | 2.23% |
| LSAT | 70.35% | 1.73% | 1.64% | **1.33**% |
| AIME | 80.84% | 45.19% | 42.39% | **41.04**% |
| MATH-500 | 64.97% | 0.94% | 0.95% | **0.93**% |
| FRAMES | 69.41% | 10.17% | 1.22% | **1.21**% |

Table 10: Routing performance on OOD queries across benchmarks reported on the CPT (85%) metric (lower is better). CPT (85%) denotes the fraction of the cost of running OpenAI o4-mini with a high reasoning budget to match 85% of its performance. Best performance is in **bold** and second best is underlined.

| Benchmark | Random-Pair | RouterBench | IRT-Router | RADAR (ours) |
|---|---|---|---|---|
| GPQA-Diamond | 74.3% | 24.53% | 31.34% | **10.86**% |
| MMLU | 63.55% | 1.96% | 1.88% | **1.73**% |
| MMLU-Redux | 63.09% | 1.99% | 1.92% | **1.85**% |
| MMLU-Pro | 74.4% | 2.44% | 2.2% | **2.13**% |
| LSAT | 70.54% | 1.73% | 1.65% | **1.5**% |
| AIME | 77.92% | — | **32.78**% | 42.9% |
| MATH-500 | 63.55% | 0.9% | 0.97% | **0.73**% |
| FRAMES | 68.74% | 16.7% | **1.21**% | 1.55% |

Table 11: Routing performance on ID queries across benchmarks reported on the CPT (95%) metric (lower is better). CPT (95%) denotes the fraction of the cost of running OpenAI o4-mini with a high reasoning budget to match 95% of its performance. Best performance is in **bold** and second best is underlined.

| Benchmark | Random-Pair | RouterBench | IRT-Router | RADAR (ours) |
|---|---|---|---|---|
| GPQA-Diamond | 89.19% | 76.86% | 76.78% | **21.1**% |
| MMLU | 88.15% | 15.27% | **6.73**% | 7.52% |
| MMLU-Redux | 86.79% | 8.69% | 5.45% | **5.16**% |
| MMLU-Pro | 91.42% | 38.78% | 43.64% | **12.08**% |
| LSAT | 90.01% | **3.14**% | 3.48% | 3.62% |
| AIME | 93.61% | 86.11% | 80.61% | **80.34**% |
| MATH-500 | 87.63% | **2.03**% | 2.26% | 2.58% |
| FRAMES | 88.84% | 77.6% | 65.76% | **31.81**% |

Table 12: Routing performance on OOD queries across benchmarks reported on the CPT (95%) metric (lower is better). CPT (95%) denotes the fraction of the cost of running OpenAI o4-mini with a high reasoning budget to match 95% of its performance. Best performance is in **bold** and second best is underlined.

| Benchmark | Random-Pair | RouterBench | IRT-Router | RADAR (ours) |
|---|---|---|---|---|
| GPQA-Diamond | 90.78% | 63.83% | 77.05% | **36.66%** |
| MMLU | 86.42% | 29.76% | 8.37% | **7.14%** |
| MMLU-Redux | 86.14% | 10.57% | 5.81% | **5.62%** |
| MMLU-Pro | 91.32% | 51.8% | 45.25% | **36.2%** |
| LSAT | 90.0% | 8.39% | **3.52%** | 5.63% |
| AIME | 92.44% | – | **77.59%** | – |
| MATH-500 | 86.34% | 2.58% | **2.13%** | 2.79% |
| FRAMES | 89.3% | – | 61.44% | **22.13%** |

Table 13: Routing performance on OOD queries across benchmarks reported on the hypervolume metric (higher is better). RADAR outperforms baselines, denoting better performance-cost tradeoffs towards the Pareto front. Best performance is in **bold** and second best is underlined.

| Benchmark | Random-Pair | RouterBench | IRT-Router | RADAR (ours) |
|---|---|---|---|---|
| GPQA-Diamond | 0.5369 | 0.7047 | 0.6938 | **0.7466** |
| MMLU | 0.6934 | 0.8398 | 0.8550 | **0.8609** |
| MMLU-Redux | 0.7298 | 0.8948 | 0.9050 | **0.9072** |
| MMLU-Pro | 0.5686 | 0.7703 | 0.7800 | **0.7858** |
| LSAT | 0.6887 | 0.9046 | **0.9175** | 0.9146 |
| AIME | 0.5283 | 0.6890 | **0.7915** | 0.7566 |
| MATH-500 | 0.7493 | 0.9326 | **0.9385** | 0.9368 |
| FRAMES | 0.6624 | 0.8230 | 0.8548 | **0.8865** |

## B.3 RESULTS ON OOD QUERIES

Table 13 and Table 14 report routing performance of all methods across 8 reasoning benchmarks evaluated in the OOD setting on the hypervolume and CPT (90%) metrics, respectively. RADAR exhibits strong query generalization capabilities.

## B.4 IMPROVING OOD PERFORMANCE ON AIME

Our analysis reveals two special characteristics of AIME. First, AIME contains queries with the greatest average difficulty, significantly higher than other benchmarks as seen in Table 15. Second, performance on AIME consistently improves with increasing reasoning length and does not plateau, unlike other benchmarks. In the OOD setting, RADAR, which is not exposed to difficult math

Table 14: Routing performance on OOD queries across benchmarks reported on the CPT (90%) metric (lower is better). CPT (90%) denotes the fraction of the cost of running OpenAI o4-mini with a high reasoning budget to match 90% of its performance. Best performance is in **bold** and second best is underlined.

| Benchmark | Random-Pair | RouterBench | IRT-Router | RADAR (ours) |
|---|---|---|---|---|
| GPQA-Diamond | 82.54% | 44.18% | 54.19% | **17.6%** |
| MMLU | 74.53% | 2.94% | **2.61%** | 2.63% |
| MMLU-Redux | 74.61% | 2.90% | 2.71% | **2.54%** |
| MMLU-Pro | 82.65% | 7.67% | 4.02% | **3.54%** |
| LSAT | 80.07% | 2.27% | **1.96%** | 2.15% |
| AIME | 84.88% | – | **55.19%** | 55.30 |
| MATH-500 | 74.94% | 1.4% | **1.29%** | 1.55% |
| FRAMES | 78.61% | 48.52% | 29.49% | **9.99%** |

Table 15: Predicted query difficulty on ID test queries comparing AIME against the remaining benchmarks. IRT difficulty values usually lie in $[-3, 3]$, with higher values indicating greater difficulty.

| Benchmark | Difficulty Avg | Difficulty Std |
|---|---|---|
| AIME | 1.038 | 1.275 |
| All Benchmarks | -0.55 | 1.018 |

Table 16: Predicted query difficulty on test queries comparing ID AIME queries to OOD AIME queries. IRT difficulty values usually lie in the range [-3, 3], with higher values indicating greater difficulty.

| Benchmark | Difficulty Avg | Difficulty Std |
|---|---|---|
| AIME (ID) | 1.038 | 1.275 |
| AIME (OOD) | -0.461 | 1.043 |

problems, underestimates query difficulty, as shown in Table 16, and underperforms by allocating less capable models.

In real-world use, it is practical to assume some exposure to math problems during model training. We perform a partial OOD experiment by exposing RADAR to a fraction of the AIME and MATH queries in the training data. In Table 17, we see a significant increase in performance with just 5% of exposure, and a similar performance to the ID setting with 30% of exposure.

## B.5 ABLATION STUDY ON SCALARIZATION

See Table 18 for an ablation study which shows Chebyshev scalarization outperforms linear scalarization on OOD queries due to its ability to explore both convex and concave points on the Pareto front.

## B.6 ABLATION STUDY ON MATRIX SIZE

See Table 19 for an ablation study on the size of the training matrix of RADAR. Using just 20% of subsampled training queries, RADAR achieves a similar performance to using the entire training set.

## B.7 ABLATION STUDY ON ADAPTIVE TESTING

We conduct an ablation study comparing our Fisher information-based adaptive testing with uniform random sampling on routing performance. Similar to well-established findings in adaptive testing (Kürşad, 2023) and online learning (Hanneke, 2014; Balcan et al., 2010) literature, we find that our method performs better and with lower variance than uniform sampling, especially when the sample size is small, and both methods converge to perform similarly as the sample size increases. We report results on ID qeuries in Table 20 (hypervolume) and Table 21 (CPT), and on OOD queries in Table 22 (hypervolume) and Table 23 (CPT).

For example, with a small sample size of just 100 items, our method consistently outperforms uniform sampling by a wide margin on ID benchmarks and on AIME and MATH benchmarks for OOD. Further, uniform sampling exhibits high variance as seen in FRAMES in the ID setting, achieving 2.6% CPT(90%) with 100 items, which counterintuitively jumps to 22.2% with a larger set of 500

Table 17: Partial OOD experimental setting showing that a small set of training queries can recover ID performance.

| Benchmark | RADAR (ID) | RADAR (OOD) | RADAR (OOD + 5%) | RADAR (OOD + 30%) |
|---|---|---|---|---|
| AIME | 0.776 | 0.7566 | 0.7665 | 0.7787 |

Table 18: Ablation study showing Chebyshev scalarization outperforms linear scalarization on OOD queries due to its ability to explore both convex and concave points on the Pareto front.

| Benchmark | Hypervolume (higher is better) | | CPT(90%) (lower is better) | |
|---|---|---|---|---|
| | RADAR (LS) | RADAR (CS) | RADAR (LS) | RADAR (CS) |
| GPQA-Diamond | 0.7280 | 0.7466 | 29.94% | 17.60% |
| MMLU | 0.8580 | 0.8609 | 2.50% | 2.63% |
| MMLU-Redux | 0.9049 | 0.9072 | 2.25% | 2.54% |
| MMLU-Pro | 0.7812 | 0.7858 | 3.81% | 3.54% |
| LSAT | 0.9165 | 0.9146 | 2.00% | 2.15% |
| AIME | 0.7464 | 0.7566 | 56.23% | 55.30% |
| MATH-500 | 0.9331 | 0.9368 | 1.41% | 1.55% |
| FRAMES | 0.8656 | 0.8865 | 21.56% | 9.99% |

Table 19: Ablation study on the size of the training matrix of RADAR. Using just 20% of subsampled training queries, RADAR achieves a similar performance to using the entire training set.

| Benchmark | Hypervolume (higher is better) | | CPT(90%) (lower is better) | |
|---|---|---|---|---|
| | RADAR (20%) | RADAR | RADAR (20%) | RADAR |
| GPQA-Diamond | 0.7526 | 0.7513 | 16.29 | 13.21 |
| MMLU | 0.8726 | 0.8720 | 2.67 | 2.69 |
| MMLU-Redux | 0.9207 | 0.9230 | 2.69 | 2.42 |
| MMLU-Pro | 0.7990 | 0.7995 | 3.59 | 3.89 |
| LSAT | 0.9175 | 0.9188 | 1.95 | 1.82 |
| AIME | 0.7832 | 0.7760 | 57.85 | 60.69 |
| MATH-500 | 0.9450 | 0.9449 | 1.15 | 1.31 |
| FRAMES | 0.8940 | 0.8762 | 10.70 | 13.11 |

items. When uniform sampling marginally performs better, its high variance might indicate a fortuitous estimate of ability rather than an accurate one. In contrast, our method provides a reliable and precise estimate of ability, which results in stable routing performance across sample sizes.

### B.8 MODEL SCALABILITY AND GENERALIZATION EVALUATION OF RADAR

We evaluate the scalability of RADAR to new RLMs by adding 8 new model configurations from the Qwen3 14B RLM. Using adaptive testing, RADAR accurately estimates the abilities of these new configurations by dynamically selecting just 5k training queries ( 12% of the training set) for

Table 20: Routing performance on ID queries across benchmarks reported on the hypervolume metric (higher is better), before (RADAR) and after (RADAR++) adding new RLM configurations from Qwen3-14B. Rnd X denotes a set of X items selected with uniform random sampling, while Fshr X denotes a set of X items selected with our method of maximum Fisher information-based adaptive testing.

| Benchmark (ID) | RADAR | RADAR++ | | | | | |
|---|---|---|---|---|---|---|---|
| | | Rnd 100 | Fshr 100 | Rnd 500 | Fshr 500 | Rnd 5000 | Fshr 5000 |
| GPQA-Diamond | 0.7513 | 0.6171 | 0.7511 | 0.7017 | 0.7513 | 0.7535 | 0.7535 |
| MMLU | 0.8609 | 0.8591 | 0.8731 | 0.8682 | 0.8720 | 0.8745 | 0.8731 |
| MMLU-Redux | 0.9072 | 0.8929 | 0.9232 | 0.9168 | 0.9230 | 0.9228 | 0.9238 |
| MMLU-Pro | 0.7858 | 0.7764 | 0.8009 | 0.7941 | 0.7995 | 0.8038 | 0.8021 |
| LSAT | 0.9146 | 0.9270 | 0.9205 | 0.9283 | 0.9188 | 0.9259 | 0.9233 |
| AIME | 0.7760 | 0.6883 | 0.7636 | 0.7311 | 0.7760 | 0.7840 | 0.7828 |
| MATH-500 | 0.9449 | 0.9426 | 0.9456 | 0.9457 | 0.9449 | 0.9478 | 0.9461 |
| FRAMES | 0.8865 | 0.8629 | 0.8763 | 0.8587 | 0.8762 | 0.8897 | 0.8830 |

Table 21: Routing performance on ID queries across benchmarks reported on the CPT (90%) metric (lower is better), before (RADAR) and after (RADAR++) adding new RLM configurations from Qwen3-14B. Rnd X denotes a set of X items selected with uniform random sampling, while Fshr X denotes a set of X items selected with our method of maximum Fisher information-based adaptive testing. NR denotes not reachable.

| Benchmark (ID) | RADAR | RADAR++ | | | | | |
| --- | --- | --- | --- | --- | --- | --- | --- |
| | | Rnd 100 | Fshr 100 | Rnd 500 | Fshr 500 | Rnd 5000 | Fshr 5000 |
| GPQA-Diamond | 13.21% | NR | 13.61% | 45.24% | 13.21% | 12.28% | 13.56% |
| MMLU | 2.63% | 2.67% | 2.69% | 2.75% | 2.69% | 2.69% | 2.69% |
| MMLU-Redux | 2.54% | 2.38% | 2.42% | 2.42% | 2.42% | 2.42% | 2.42% |
| MMLU-Pro | 3.54% | 3.73% | 3.9% | 3.93% | 3.89% | 3.9% | 3.9% |
| LSAT | 2.15% | 1.82% | 1.82% | 1.82% | 1.82% | 1.82% | 1.82% |
| AIME | 60.69% | NR | 65.12% | 72.43% | 60.69% | 57.7% | 58.45% |
| MATH-500 | 1.31% | 1.34% | 1.31% | 1.32% | 1.31% | 1.32% | 1.31% |
| FRAMES | 9.99% | 2.62% | 13.06% | 22.20% | 13.11% | 3.25% | 5.97% |

Table 22: Routing performance on OOD queries across benchmarks reported on the hypervolume metric (higher is better), before (RADAR) and after (RADAR++) adding new RLM configurations from Qwen3-14B. Rnd X denotes a set of X items selected with uniform random sampling, while Fshr X denotes a set of X items selected with our method of maximum Fisher information-based adaptive testing.

| Benchmark (OOD) | RADAR | RADAR++ | | | | | |
| --- | --- | --- | --- | --- | --- | --- | --- |
| | | Rnd 100 | Fshr 100 | Rnd 500 | Fshr 500 | Rnd 5000 | Fshr 5000 |
| GPQA-Diamond | 0.7466 | 0.7402 | 0.6362 | 0.7466 | 0.7177 | 0.7467 | 0.7463 |
| MMLU | 0.8609 | 0.87 | 0.8679 | 0.8697 | 0.8684 | 0.8697 | 0.8698 |
| MMLU-Redux | 0.9072 | 0.9086 | 0.9093 | 0.9090 | 0.9098 | 0.9094 | 0.9091 |
| MMLU-Pro | 0.7858 | 0.7966 | 0.7927 | 0.7958 | 0.7928 | 0.7961 | 0.7951 |
| LSAT | 0.9146 | 0.9287 | 0.9273 | 0.9264 | 0.9267 | 0.9278 | 0.9255 |
| AIME | 0.7566 | 0.6543 | 0.7566 | 0.7385 | 0.7566 | 0.7694 | 0.7566 |
| MATH-500 | 0.9368 | 0.9186 | 0.9368 | 0.9367 | 0.9368 | 0.9389 | 0.9368 |
| FRAMES | 0.8865 | 0.8658 | 0.8699 | 0.8864 | 0.8872 | 0.8938 | 0.8931 |

Table 23: Routing performance on OOD queries across benchmarks reported on the CPT (90%) metric (lower is better), before (RADAR) and after (RADAR++) adding new RLM configurations from Qwen3-14B. Rnd X denotes a set of X items selected with uniform random sampling, while Fshr X denotes a set of X items selected with our method of maximum Fisher information-based adaptive testing. NR denotes not reachable.

| Benchmark (OOD) | RADAR | RADAR++ | | | | | |
| --- | --- | --- | --- | --- | --- | --- | --- |
| | | Rnd 100 | Fshr 100 | Rnd 500 | Fshr 500 | Rnd 5000 | Fshr 5000 |
| GPQA-Diamond | 17.6% | 22.6% | NR | 15.8% | 37% | 17.48% | 16.63% |
| MMLU | 2.63% | 2.63% | 2.63% | 2.63% | 2.63% | 2.63% | 2.63% |
| MMLU-Redux | 2.54% | 2.54% | 2.54% | 2.54% | 2.54% | 2.54% | 2.54% |
| MMLU-Pro | 3.54% | 3.69% | 3.54% | 3.69% | 3.54% | 3.7% | 3.56% |
| LSAT | 2.15% | 2.19% | 2.15% | 2.53% | 2.15% | 2.48% | 2.15% |
| AIME | 55.3% | 59.2% | 55.3% | 56.5% | 55.3% | 52.5% | 55.3% |
| MATH-500 | 1.55% | 1.98% | 1.55% | 1.53% | 1.55% | 1.57% | 1.55% |
| FRAMES | 9.99% | 8.72% | 2.58% | 10% | 5.16% | 2.57% | 5.25% |

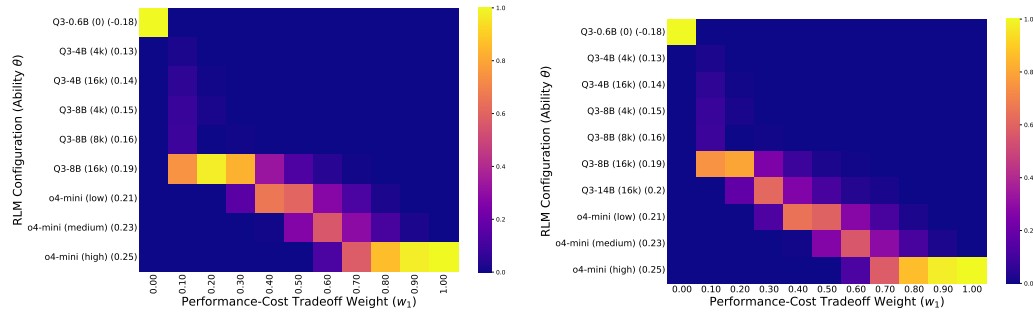

Figure 4: Fraction of routing calls on OOD queries from FRAMES spread across RLM configurations when varying the performance-cost tradeoff weight before (left) and after (right) adding new RLM configurations from Qwen3-14B. RADAR rapidly estimates the ability of Qwen3-14B at 16K reasoning budget to leverage it for improved performance.

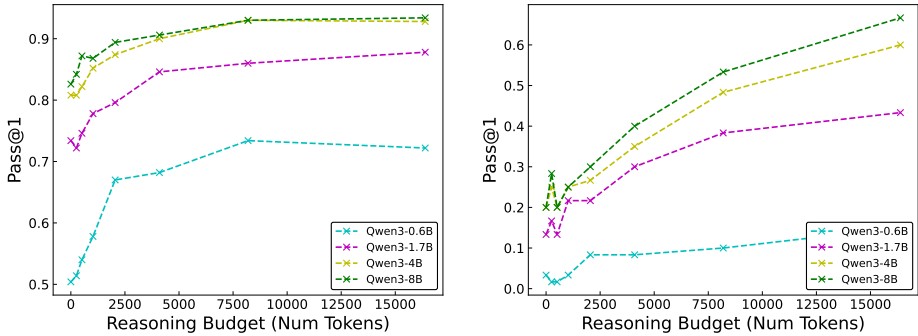

Figure 5: **Left**: Our pilot study on MATH-500 (Hendrycks et al., 2021c) shows a performance differential over (RLM, reasoning budget) configurations with the smallest RLM already solving over $50\%$ of the queries with minimal reasoning. **Right**: Performance on AIME consistently increases with an increase in model size and reasoning budget.

evaluation, resulting in improved routing performance. Figure 4 shows how routing shifts to new RLM configurations.

## B.9 THROUGHPUT ANALYSIS

For a throughput analysis, we compute the queries/second for the smallest (0.6B) and largest (8B) open-source Qwen3 models on the MATH-500 benchmark, averaged across three runs on a single Nvidia A100 80GB GPU, using vLLM for batched inference and embedding. We exclude OpenAI o4-mini models due to their variable API-based throughput. Qwen3-0.6B with zero reasoning budget processes queries at 1.1529 queries/sec, while Qwen3-8B with 16K reasoning budget processes queries at 0.2337 queries/sec. Adding RADAR's routing overhead decreases throughput by 0.78% for Qwen3-0.6B with zero reasoning budget, to 1.1438 queries/sec, and by 0.15% for Qwen3-8B with a 16K reasoning budget, to 0.2237 queries/sec. Depending on the user-specified performance-cost tradeoff weight, RADAR routes queries to a convex combination of model configurations, with the throughput therefore bounded between 1.1438 queries/sec as the upper bound and 0.2237 queries/sec as the lower bound. Multi-GPU implementation and inference optimization methods can further improve RADAR's throughput.

## B.10 PERFORMANCE VS REASONING BUDGET CURVES

We include performance vs reasoning budget curves in Figure 5.

## B.11 PERFORMANCE-COST PARETO CURVES

We show Pareto performance-cost tradeoff curves for all methods on ID queries across all benchmarks in Figure 6, and on OOD queries across all benchmarks in Figure 7.

# C DATASET DESCRIPTION

The 9 benchmarks are: 1) **AIME** (MAA, 2024): A benchmark of competition math problems from American Invitational Mathematics Examination (AIME), which determines qualification for the United States Mathematical Olympiad, 2) **MATH** (Hendrycks et al., 2021c): A benchmark of math problems drawn from various math competitions, 3) **GPQA** (Rein et al., 2024): A benchmark of PhD-level science multiple-choice questions (MCQs) written by domain experts, 4) **LSAT** (Wang et al., 2022; Zhong et al., 2021): A benchmark of MCQs from the three tasks of the Law School Admission Test (LSAT), including analytical reasoning, logical reasoning and reading comprehension, 5) **MMLU** (Hendrycks et al., 2021b;a): A benchmark of MCQs from various branches of knowledge covering diverse domains, 6) **MMLU Redux** (Gema et al., 2024): A subset of MMLU with manually corrected MCQs to remove errors from the original benchmark, 7) **MMLU Pro** (Wang et al., 2024): An enhanced MMLU benchmark with a focus on reasoning questions with increased answer options from 4 to 10, 8) **DROP** (Dua et al., 2019): A benchmark of reading comprehension questions requiring discrete reasoning over the question's associated paragraph, and 9) **FRAMES** (Krishna et al., 2024): A benchmark of long-context reasoning-based questions associated with multiple wikipedia articles. Table 4 shows the statistics of each dataset.

## C.1 PREPROCESSING DETAILS

Across datasets, we standardize formatting; compute prompt token counts with the `Qwen/Qwen3-0.6B` tokenizer (no padding, truncation, or added special tokens); and discard items exceeding a configured token budget. To prevent leakage, we compute a content-based item key and apply deduplication when specified for a given dataset, with some datasets deferring duplicate handling to later analysis. Where applicable, we normalize available metadata and extract missing numeric answers. All datasets are mapped into a unified prompt–response format; detailed prompt templates are provided in Appendix D.4.

**AIME.** We preprocess AIME by standardizing sources and prompts, then filtering and deduplicating. Training data span years 1983–2023,[2] while test data consist of the union of unique items from AIME 2024[3] and 2025[4] to reduce evaluation variance. Evaluating on AIME 2024 and AIME 2025 separately resulted in high evaluation variance even after averaging over multiple runs. Examples with prompt length exceeding the maximum token budget are discarded. For AIME 2025, only the problem text and numeric answer are retained; missing fields such as solution or difficulty are set to "NA."

**MATH.** We construct the training split by combining the seven subject configurations of the MATH dataset[5] (7,500 problems total) and use the fixed 500-problem test set.[6] Examples exceeding the maximum prompt length are removed. When an explicit numeric answer is missing, it is extracted from the provided solution. Metadata such as subject/type and level are normalized, and any available `unique_id` is preserved. We shuffle the data with a fixed seed.

**GPQA.** We preprocess GPQA[7] by combining the main and diamond subsets and verifying that diamond IDs are contained within the main set. Each example is reformatted into a multiple-choice format with options A–D, and the correct answer is recorded as a letter. Answer options are randomly permuted with a fixed seed. Items exceeding the maximum prompt length are filtered out.

---

[2]https://github.com/rllm-org/rllm/blob/deepscaler/deepscaler/data/train/aime.json
[3]https://github.com/rllm-org/rllm/blob/deepscaler/deepscaler/data/test/aime.json
[4]https://huggingface.co/datasets/yentinglin/aime_2025
[5]https://huggingface.co/datasets/HuggingFaceH4/MATH/viewer
[6]https://huggingface.co/datasets/HuggingFaceH4/MATH-500
[7]https://huggingface.co/datasets/Idavidrein/gpqa

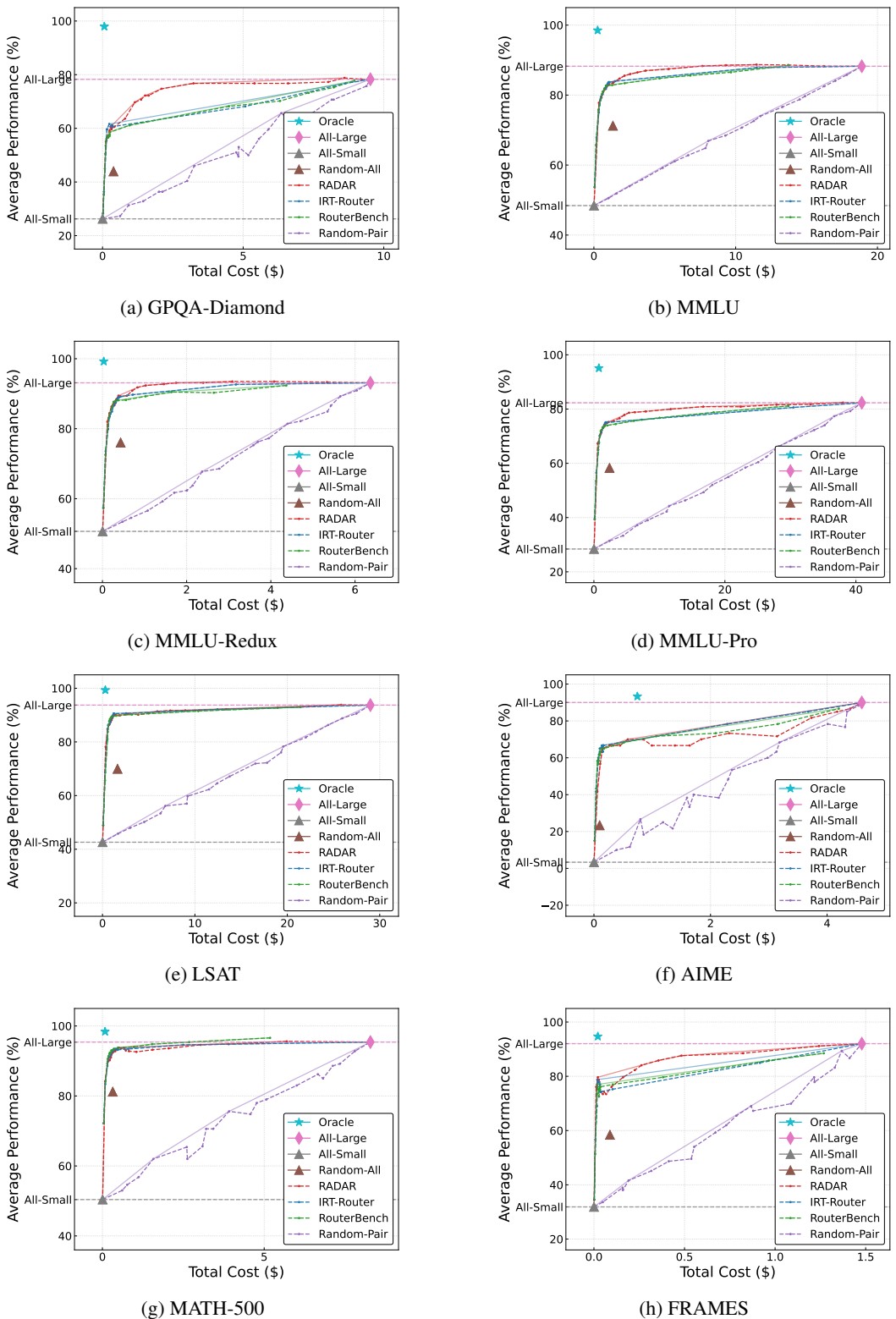

Figure 6: We show the Pareto performance-cost tradeoff curves for all methods on ID queries across benchmarks. RADAR outperforms baselines, denoting better performance-cost tradeoffs towards the Pareto front. Solid lines in the figure denote the convex hull corresponding to each curve.

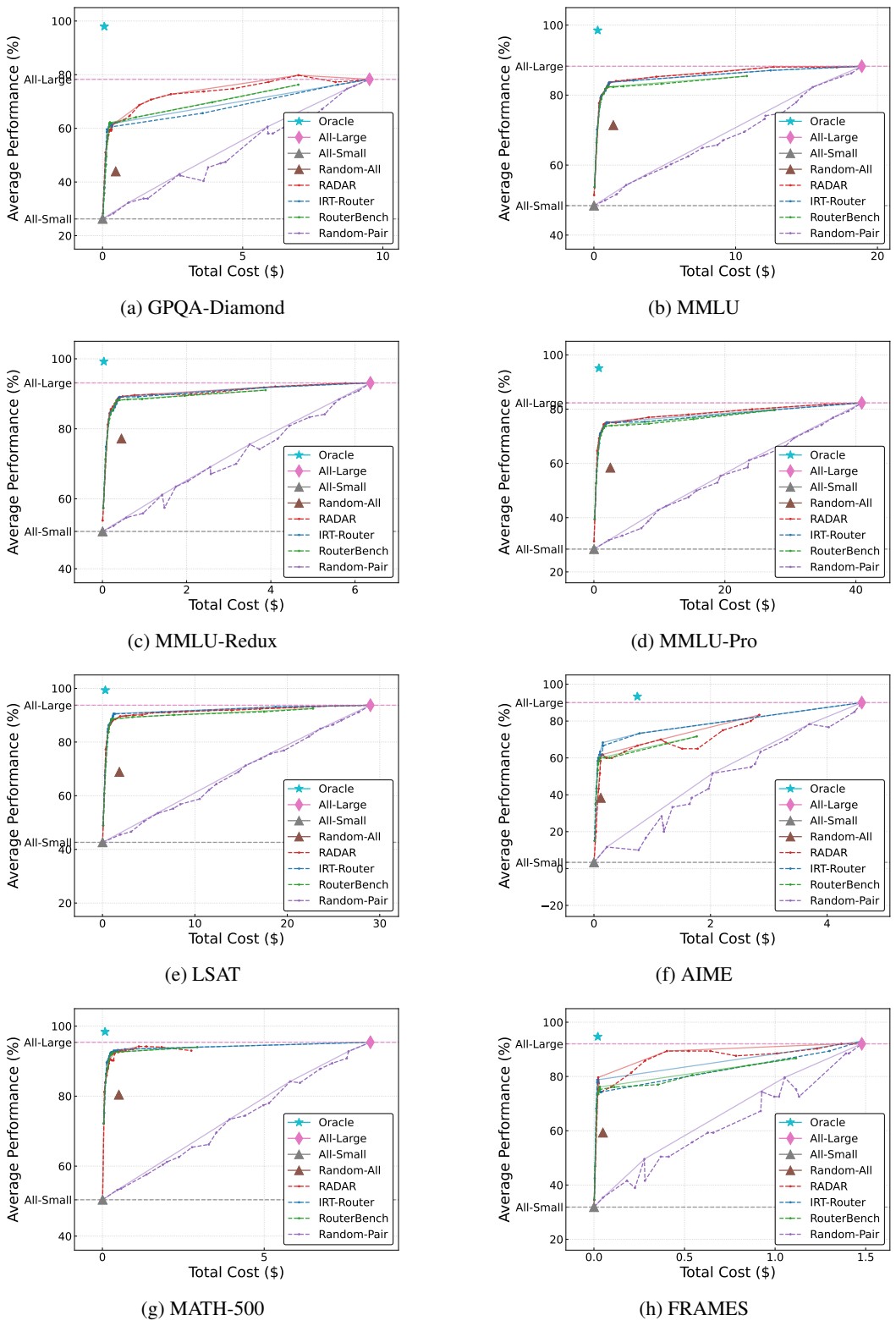

Figure 7: We show the Pareto performance-cost tradeoff curves for all methods on OOD queries across benchmarks. RADAR outperforms baselines, denoting better performance-cost tradeoffs towards the Pareto front. Solid lines in the figure denote the convex hull corresponding to each curve.

Deduplication is performed using a content-based key, and evaluation is conducted with the diamond subset as the test set.

**LSAT.** We preprocess LSAT by standardizing items from the official AR-LSAT release,[8] spanning reading comprehension, logical reasoning, and analytical reasoning. Each example is reformatted into a multiple-choice prompt with options A–E. Items exceeding the token budget are discarded. We preserve the original section and split labels, and record the gold answer both as an index and as a letter. Data are shuffled deterministically with a fixed seed.

**MMLU.** We use only the official test split of MMLU.[9] Each example is converted into a standardized multiple-choice prompt (options A–D), retaining both the textual correct answer and its letter index. Items exceeding the token threshold are discarded. Subject metadata are preserved, and the dataset is shuffled with a fixed seed.

**MMLU Pro.** We use the public test split of MMLU Pro.[10] Each example is constructed into a multiple-choice prompt with options A–J, and the gold answer is stored as a letter. Prompts exceeding a 32k token budget are discarded. The dataset is shuffled deterministically with a fixed seed.

**MMLU Redux.** We preprocess MMLU Redux[11] by aggregating all subject configurations and discarding items flagged with metadata $\text{error\_type} \neq \text{ok}$. Each example is normalized and converted into a multiple-choice prompt (options A–D), with both the correct answer letter and text recorded. Items longer than the token budget are removed, and the dataset is shuffled deterministically with a fixed seed.

**FRAMES.** We preprocess FRAMES[12] by retrieving and cleaning the corresponding Wikipedia pages for each example. Cleaning removes site chrome, images, hyperlinks, citation markers, and irrelevant sections, while preserving tables and converting text to Markdown. Examples are then converted into a document QA style format. Items exceeding the token budget are filtered out, and unique article texts are cached to avoid re-downloading.

# D ADDITIONAL EXPERIMENTAL DETAILS

## D.1 EVALUATION SETUP AND IMPLEMENTATION DETAILS.

We conduct both in-distribution (ID) and out-of-distribution (OOD) evaluations. For ID experiments, we aggregate the training splits of all 8 benchmarks into a single training set for training the 2PL IRT model (see Section 3.3) and report performance on the test split of each benchmark separately. We use an $80\% - 20\%$ train-test split for benchmarks without a predefined test set. For OOD, for each benchmark, we aggregate the training splits of the other remaining *non-overlapping* benchmarks into a single training set and report performance on the test split of this benchmark. For example, for the OOD experiment on AIME, the training split of AIME and of overlapping benchmarks (MATH since MATH includes questions from AIME), are held out from the training set. We route over 35 configurations comprising OpenAI **o4-mini** (budgets: low, medium, high) and **Qwen3** models (0.6B/1.7B/4B/8B) with budgets 0, 256, 512, 1k, 2k, 4k, 8k, 16k (Yang et al., 2025; OpenAI, 2025). Each configuration is evaluated once per training query with standard prompts (Appx. D.4); for AIME's small test set, we average over eight runs. In total, we collected 1.75 million binary responses over 50,139 unique questions across train/test splits of all eight benchmarks.

---

[8] https://github.com/zhongwanjun/AR-LSAT/tree/main/complete_lsat_data
[9] https://huggingface.co/datasets/cais/mmlu
[10] https://huggingface.co/datasets/TIGER-Lab/MMLU-Pro
[11] https://huggingface.co/datasets/edinburgh-dawg/mmlu-redux-2.0
[12] https://huggingface.co/datasets/google/frames-benchmark

## D.2 HARDWARE

For all open-source models, we use vLLM (Kwon et al., 2023) to host the model. All experiments involving open-source models are run on NVIDIA A100 80GB GPUs. Each model is hosted using a single such GPU.

## D.3 BASELINES

For **RouterBench** (Hu et al., 2024), we adopt its k-nearest neighbors (kNN) parameterization, which performs best (Chen et al., 2025). We adapt a concurrent IRT-based model routing work, **IRT-Router** (Song et al., 2025), by using the same 2PL IRT model parameterization and query embedder as RADAR, thereby improving its embedder to handle long-context queries for fairness. For both model routing methods, **RouterBench** and **IRT-Router**, we adapt them to RLMs by using all RLMs at their respective fixed maximum budgets, and use their performance-cost formulation similar to linear scalarization (see Equation 3). In addition, we include simple heuristic-based baselines: **All-Large** (o4-mini at high budget) and **All-Small** (Qwen3 0.6B at zero budget) as approximate upper and lower bounds on performance and cost, respectively. The **Oracle** router, provided with model configuration performance on test queries, serves as an idealized approximate upper bound of the performance-cost tradeoff by picking the cheapest best-performing configuration. **Random-All** serves as a diversity baseline, selecting a configuration at random to answer each query. **Random-Pair** selects the largest configuration (o4-mini at high budget) with probability $w_1$, and the smallest configuration (Qwen3 0.6B at zero budget) with probability $1 - w_1$, where $w_1$ is the user-defined performance-cost tradeoff weight.

## D.4 QA PROMPTS

The prompts below are applied to all RLM configurations.

For AIME and MATH, we use the following prompt:

```
{question}
Please reason step by step, and put your final answer within \boxed{}.
```

For GPQA, LSAT, MMLU, and MMLU Redux, we use the following prompt:

```
Answer the following multiple choice question.

{question}

A) {option_A}
B) {option_B}
C) {option_C}
D) {option_D}

Please reason step by step, and put your final answer option within \boxed{}.
Only put the letter in the box, e.g. \boxed{A}. There is only one correct
answer.
```

For MMLU Pro, we use the following prompt:

```
Answer the following multiple choice question.

{question}

{options}

Please reason step by step, and put your final answer option within \boxed{}.
Only put the letter in the box, e.g. \boxed{A}. There is only one correct
answer.
```

For FRAMES, we assemble the prompt programmatically that includes the context of all documents relevant to the question:

```
1  prompt = f"""You are asked to read {len(docs)} Wikipedia article extracts
2  and answer a question. Please reason step by step, and put your final
3  answer within \\boxed{}."""
4  for i, doc in enumerate(docs):
5      prompt += f"\n\n# Wikipedia article {i+1}:\n{doc}"
6  prompt += f"\n\n# Question: {question}"
7  prompt += """\n\nPlease reason step by step, and put your final answer within
       ↪ \\boxed{}."""
```

### D.5 IRT IMPLEMENTATION DETAILS IN RADAR

We employ a two-parameter logistic (2PL) IRT model implemented as a custom PyTorch model class. Input queries are first processed into fixed embeddings by a frozen-weight `Qwen/Qwen3-Embedding-8B`; the dimension $d_q = 4096$ for both the query embedding and the learnable weights $\boldsymbol{w}_a, \boldsymbol{w}_b$. Training runs for $100$ epochs with learning rate $5 \times 10^{-4}$, batch size $32$ for both training and evaluation, gradient clipping at norm $1.0$, and gradient accumulation of $1$ step.

### D.6 METRICS

Our formulation of adaptive reasoning as an MOO naturally lends the use of the **hypervolume** indicator metric (Emmerich & Deutz, 2018), which measures the size of the dominated space recovered by the MOO solution method, with a higher value indicating performance close to the Pareto front. In our two-dimensional routing MOO, hypervolume intuitively measures the area under the performance-cost tradeoff curve recovered by the routing method for various values of tradeoff weights $w_1$. An advantage of hypervolume over similar area-based metrics defined in existing routing work (e.g., AIQ in Hu et al. (2024)) is its generalizability to measuring the performance of a multi-dimensional routing MOO. In future work, additional dimensions such as latency, bias, and carbon emissions can be added to the routing MOO. We also formulate a **cost-performance threshold (CPT)** metric, similar to the call-performance threshold metrics in Ong et al. (2024), a useful metric for real-world applications quantifying the cost required to reach a specified performance level. Given a performance threshold $x\%$, CPT($x\%$) measures the minimum cost required to achieve $x\%$ of the performance of the largest configuration (OpenAI o4-mini with high reasoning budget). We normalize this cost to $[0, 1]$ by dividing by the cost of running the largest configuration. Therefore, a CPT($90\%$) of $0.1$ implies that the routing method can match $90\%$ of the performance of o4-mini high at $10\%$ of its cost.

## E THE USE OF LLMS FOR THIS PAPER

LLM usage is limited to writing and editing suggestions, such as word and phrase choices, highlighting grammar issues in the authors' draft, and polishing the writing. Other LLM usage includes LLM-augmented search engines to assist in identifying prior and concurrent related work.

