# OpenReview forum: "RADAR: Reasoning-Ability and Difficulty-Aware Routing for Reasoning LLMs"
_ICLR.cc/2026/Conference — ICLR 2026 Poster_

### Official Review · Reviewer_LEBC · 2025-10-28

**Soundness:** 3
**Presentation:** 3
**Contribution:** 3
**Rating:** 8
**Confidence:** 3

**Summary:**

This work considers the problem of query-based routing reasoning language models (RLM) with variable budget.
The authors here formulate such problems into multi-objective optimization (MOO) paradigm with certain scalarization (linear/Chebyshev) for both performance and cost.
The MOO's variables include 1) model-budget configuration and 2) query difficulty.
Model-budget configurations are (model, token) pairs with reasoning token count discretized and the authors propose to leverage IRT (item response theory) to learn the difficulty/discrimination projector with a frozen embedding model and a model-configuration _ability_ parameter to maintain both interpretability (low parameter) and low latency during inference.

This paper is well written and the idea is well presented with good motivation and strong results.

**Strengths:**

1. Well-motivated: authors propose a method to achieve pareto frontier in terms of performance vs. cost in the case of multiple RLMs with varying size and reasoning budgets. This is a very realistic scenario and could be rather impactful. In addition, the proposed method is easy to adapt to any sort of models as they are treated as black boxes and readily extensible to newer and hopefully stronger or faster models.
2. Good performance: the authors compare RADAR against IRT Router and RouterBench and a couple of heuristics and show that RADAR achieves better hypervolume metric.
3. Fast: for each inference (unseen) query, RADAR performs an embedding step through an embedding model and a proposed linear projector to get the difficulty/discrimination scalars and feed these 2 scalars into the IRT model. This is rather fast in comparison to the later pipeline considering the number of parameters and reasoning budget of the following models.
4. Interpretability: IRT model only has two parameters (difficulty & discrimination in terms of input query) which affords natural interpretability.

**Weaknesses:**

The IRT model used in the paper assumes monotonicity, and the projectors are linear with the input embedding model frozen. I know it is probable that using a more complex model could result in a mere marginal improvement in terms of hypervolume, but would like to see some results in this regard. So is the case with the IRT model

**Questions:**

Same as weaknesses, would love to see some experiments on some more capable difficulty/discrimination projectors or a bit more complex model than IRT. While it is very reasonable and probable to end up with the conclusion that these more complicated model could obfuscate the interpretability and spike up latency while yielding only marginal performance, it will further strengthen the argument for RADAR.

---

> ### Author Response · Authors · 2025-11-25
> **Response to Reviewer LEBC**
>
> Thank you for your thorough review and insightful questions, and for highlighting the novelty and effectiveness of our method.
>
>
>
> **Ablation Study on Linear Transform for Query Difficulty**
>
> Our design choice of simple, interpretable linear transformations and a frozen embedding model helps ensure a minimal router latency of around 7 milliseconds. We agree that more complicated architectures could better model the difficulty of queries. However, our goal in RADAR is not to find the best architecture but to present an easily customizable MOO-grounded routing framework with design choices including base IRT models, transformations for difficulty, MOO solution techniques, among others.
>
> We test the sufficiency of our linear transformation for query difficulty in two ways. First, we performed a case study on MATH-500 where queries are annotated with one of five levels of increasing difficulty, one of the few datasets with real problem difficulty annotations. We find a moderately positive Pearson correlation coefficient of 0.509 between RADAR-estimated query difficulties and the five ground-truth levels, showing that RADAR is able to capture real question difficulty. We have reported our findings in Section 4.3.
>
> Second, we perform a new ablation study with a classic two-layer multilayer perceptron (MLP) model using a ReLU non-linearity to obtain the difficulty and discrimination of queries. Our results on the hypervolume metric show similar performance in both the ID and OOD experimental setting across benchmarks. On AIME and MATH-500, MLP is slightly worse in the ID setting, and sbetter in the OOD setting. These none to marginal gains, obtained at the cost of diminished interpretability and increased latency, further justify the simplicity of our design choices in RADAR.
>
>
> Table 1: Routing performance on ID queries across benchmarks reported on the hypervolume metric (higher is better).
> | Benchmark (ID) | RADAR | RADAR (MLP) |
> | - | - | - |
> | GPQA-Diamond | 0.7513 | 0.7308 |
> | MMLU | 0.8720 | 0.8707 |
> | MMLU-Redux | 0.9230 | 0.9239 |
> | MMLU-Pro | 0.7995 | 0.7955 |
> | LSAT | 0.9188 | 0.9132 |
> | AIME | 0.7760 | 0.7687 |
> | MATH-500 | 0.9449 | 0.9365 |
> | FRAMES | 0.8762 | 0.8777 |
>
>
> Table 2: Routing performance on ID queries across benchmarks reported on the CPT (90%) metric (lower is better).
> | Benchmark (ID) | RADAR | RADAR (MLP) |
> | - | - | - |
> | GPQA-Diamond | 13.21% | 14.11% |
> | MMLU | 2.69% | 2.8% |
> | MMLU-Redux | 2.42% | 2.51% |
> | MMLU-Pro | 3.89% | 3.91% |
> | LSAT | 1.82% | 2.32% |
> | AIME | 60.69% | 53.28% |
> | MATH-500 | 1.31% | 1.33% |
> | FRAMES | 13.11% | 11.37% |
>
>
> Table 3: Routing performance on OOD queries across benchmarks reported on the hypervolume metric (higher is better).
> | Benchmark (OOD) | RADAR | RADAR (MLP) |
> | - | - | - |
> | GPQA-Diamond | 0.7466 | 0.7245 |
> | MMLU | 0.8609 | 0.8573 |
> | MMLU-Redux | 0.9072 | 0.9067 |
> | MMLU-Pro | 0.7858 | 0.7844 |
> | LSAT | 0.9146 | 0.9101 |
> | AIME | 0.7566 | 0.7726 |
> | MATH-500 | 0.9368 | 0.9424 |
> | FRAMES | 0.8865 | 0.8771 |
>
>
> Table 4: Routing performance on OOD queries across benchmarks reported on the CPT (90%) metric (lower is better).
> | Benchmark (OOD) | RADAR | RADAR (MLP) |
> | - | - | - |
> | GPQA-Diamond | 17.6% | 23.49% |
> | MMLU | 2.63% | 2.82% |
> | MMLU-Redux | 2.54% | 2.63% |
> | MMLU-Pro | 3.54% | 4.82% |
> | LSAT | 2.15% | 2.12% |
> | AIME | 55.30% | 59% |
> | MATH-500 | 1.55% | 1.46% |
> | FRAMES | 9.99% | 11.17% |

---

### Official Review · Reviewer_EkVW · 2025-10-31

**Soundness:** 3
**Presentation:** 3
**Contribution:** 2
**Rating:** 4
**Confidence:** 3

**Summary:**

This paper introduces RADAR (Reasoning–Ability and Difficulty-Aware Routing), a lightweight and interpretable routing framework that optimizes the trade-off between performance and cost when deploying reasoning language models (RLMs) with varying model sizes and reasoning budgets.
RADAR draws inspiration from Item Response Theory (IRT) in psychometrics to estimate query difficulty and model-budget ability. Using these interpretable parameters, it routes easier queries to cheaper, smaller models and harder ones to more capable configurations. The authors formalize routing as a multi-objective optimization (MOO) problem balancing performance and cost via linear and Chebyshev scalarization.
Experiments across 8 reasoning benchmarks (AIME, MATH-500, GPQA, LSAT, MMLU, MMLU-Redux, MMLU-Pro, FRAMES) demonstrate that RADAR achieves superior performance-cost Pareto efficiency, generalizes well to out-of-distribution queries, and scales effectively when adding new models. It operates in real-time with ~7 ms routing latency.

**Strengths:**

First to frame reasoning model routing as a Multi-Objective Optimization (MOO) problem. Innovative use of Item Response Theory for modeling reasoning ability and query difficulty, an elegant and interpretable alternative to opaque routing regressors.

The methodology is rigorous, combining psychometric modeling with MOO and cost modeling in a consistent mathematical framework.
Experiments are extensive, covering 8 benchmarks and both in-distribution (ID) and out-of-distribution (OOD) settings, with clear comparisons to baselines such as RouterBench and IRT-Router.

The paper is very well written and structured, with clear motivation.

The work offers a generalizable and practical approach for efficient LLM deployment under cost constraints.

**Weaknesses:**

1. While RADAR clearly outperforms IRT-Router and RouterBench, the paper does not compare against mixture-of-experts or test-time adaptation methods that could serve as strong baselines for dynamic routing under cost constraints.
2. More discussion on relation to cascading methods (e.g., FrugalGPT) would strengthen the positioning.
3. Although latency is reported (~7 ms), no end-to-end system-level throughput analysis is provided.
4. Some OOD performance dips (e.g., AIME benchmark) suggest the need for improved handling of unseen high-difficulty queries.
5. The model for estimating difficulty is too simple and may not capture the real question difficulty.

**Questions:**

Check weakness above.
1. A comparison/discussion with previous works on selecting the best answer from different LLMs would have been useful for latency.
i. Uncertainty-Aware Answer Selection for Improved Reasoning in Multi-LLM Systems.
ii. Scalable best-of-n selection for large language models via self-certainty.
2. Which embedding model was used to represent queries?
3. Could RADAR dynamically adjust the user weight w₁ during a session (based on feedback or budget exhaustion) to continuously optimize global cost-performance?

---

> ### Author Response · Authors · 2025-11-25
> **Response to Reviewer EkVW (Part 1)**
>
> Thank you for your thorough review and insightful questions, and for highlighting the novelty and effectiveness of our method.
>
>
> **Comparison with Mixture-of-experts and Test-time Adaptation Methods**
>
> Our approach is complementary to these methods and can be used in conjunction with them. Specifically, although state-of-the-art open and closed models already use mixture-of-experts architectures, they have different strenghts and weaknesses, and do not dominate other models on all existing benchmarks. In such settings, post-hoc model routing can be used to leverage the streghts of multiple models instead of relying on a single one.
>
>
> **Discussion of Cascade-based Routing Methods**
>
> Our method makes a single routing decision before any model processes the query. In contrast, cascades (e.g., frugalGPT) make dynamic mid-inference decisions, which can trigger model switches, lose KV-cache benefits, and force full reprocessing which is especially costly for long docQA queries. As a result, cascades incur higher inference-time overhead and are less plug-and-play with existing models.
>
> Our design choices in RADAR are a deliberate trade-off between routing complexity, inference-time compute, and accuracy. RADAR is simpler and faster (routing adds 7ms per query in our setup), though potentially less accurate than sophisticated cascades. Our hypervolume performance shows we are close to Pareto-optimal. The marginal accuracy gains from cascading would not justify the extra latency and engineering complexity. Further, these two routing-approaches can be combined in a hybrid design for future work. RADAR can seed a cascade with a strong initial model to reduce mid-generation model switches and total latency.
>
>
>
> **Throughput Analysis**
>
> For a throughput analysis we compute the queries/second of the smallest (0.6B) and largest (8B) open source Qwen3 models on the MATH-500 benchmark averaged across 3 runs on a single Nvidia A100 80GB GPU using vLLM for batched inference and embedding. We exclude OpenAI o4-mini models due to their variable API-based throughput. Qwen3-0.6B with zero reasoning budget processes queries at 1.1529 queries/sec while Qwen3-8B with 16K reasoning budget processes queries at 0.2337 queries/sec. Adding RADAR's routing overhead decreases throughput by 0.78% for Qwen3-0.6B with zero reasoning budget to 1.1438 queries/sec and by 0.15% for Qwen3-8B with a 16K reasoning budget to 0.2237 queries/sec. Depending on the user specified performance-cost tradeoff weight, RADAR routes queries to a convex combination of model configurations with the throughput therefore bounded by 1.1438 queries/sec as the upper bound and 0.2237 queries/sec as the lower bound. Multi-GPU implementation and inference optimization methods can further improve the throughput of RADAR.
>
>
>
> **Improving OOD Performance on AIME**
>
> RADAR exhibits strong generalization performance to OOD queries consistently across benchmarks with a small decrease in performance compared with in-distribution queries even on long-context docQA. The only benchmark with a slight decrease in performance is AIME. Our analysis reveals two special characteristics of AIME. First, AIME contains queries with the highest average difficulty, significantly higher than other benchmarks. Second, performance on AIME consistently improves with an increase in reasoning length and does not plateau unlike other benchmarks. In the OOD setting, RADAR which is not exposed to difficult math problems, underestimates query difficulty and underperforms by allocating less capable models.
>
>
> Table 1: Predicted query difficulty on test questions. IRT difficulty values usually lie in [-3, 3] with a higher value denoting higher difficulty.
> | Benchmark | Diff Avg | Diff Std |
> | - | - | - |
> | AIME (ID) | 1.038 | 1.275 |
> | AIME (OOD)| -0.461 | 1.043 |
>
>
> For real-world usage, it is practical to assume some model training exposure to math problems. We perform a partial OOD experiment where we expose RADAR to a fraction of AIME and MATH queries in the training data. We see a significant increase in performance with just 5% of exposure, and a similar performance to the ID setting with 30% of exposure.
>
>
> Table 2: Partial OOD experimental setting showing a small set of training queries can recover ID performance.
> | Benchmark | RADAR (ID) | RADAR (OOD) | RADAR (OOD + 5%) | RADAR (OOD + 30%) |
> | - | - | - | - | - |
> | AIME | 0.776 | 0.7566 | 0.7665 | 0.7787 |

---

> > ### Author Response · Authors · 2025-11-25
> > **Response to Reviewer EkVW (Part 2)**
> >
> > **Ablation Study on Linear Transform for Query Difficulty**
> >
> > Our design choice of simple, interpretable linear transformations and a frozen embedding model helps ensure a minimal router latency of around 7 milliseconds. We agree that more complicated architectures could better model the difficulty of queries. However, our goal in RADAR is not to find the best architecture but to present an easily customizable MOO-grounded routing framework with design choices including base IRT models, transformations for difficulty, MOO solution techniques, among others.
> >
> > We test the sufficiency of our linear transformation for query difficulty in two ways. First, we performed a case study on MATH-500 where queries are annotated with one of five levels of increasing difficulty, one of the few datasets with real problem difficulty annotations. We find a moderately positive Pearson correlation coefficient of 0.509 between RADAR-estimated query difficulties and the five ground-truth levels, showing that RADAR is able to capture real question difficulty. We have reported our findings in Section 4.3.
> >
> > Second, we perform a new ablation study with a classic two-layer multilayer perceptron (MLP) model using a ReLU non-linearity to obtain the difficulty and discrimination of queries. Our results on the hypervolume metric show similar performance in both the ID and OOD experimental setting across benchmarks. On AIME and MATH-500, MLP is slightly worse in the ID setting, and sbetter in the OOD setting. These none to marginal gains, obtained at the cost of diminished interpretability and increased latency, further justify the simplicity of our design choices in RADAR.
> >
> >
> > Table 3: Routing performance on ID queries across benchmarks reported on the hypervolume metric (higher is better).
> > | Benchmark (ID) | RADAR | RADAR (MLP) |
> > | - | - | - |
> > | GPQA-Diamond | 0.7513 | 0.7308 |
> > | MMLU | 0.8720 | 0.8707 |
> > | MMLU-Redux | 0.9230 | 0.9239 |
> > | MMLU-Pro | 0.7995 | 0.7955 |
> > | LSAT | 0.9188 | 0.9132 |
> > | AIME | 0.7760 | 0.7687 |
> > | MATH-500 | 0.9449 | 0.9365 |
> > | FRAMES | 0.8762 | 0.8777 |
> >
> >
> > Table 4: Routing performance on ID queries across benchmarks reported on the CPT (90%) metric (lower is better).
> > | Benchmark (ID) | RADAR | RADAR (MLP) |
> > | - | - | - |
> > | GPQA-Diamond | 13.21% | 14.11% |
> > | MMLU | 2.69% | 2.8% |
> > | MMLU-Redux | 2.42% | 2.51% |
> > | MMLU-Pro | 3.89% | 3.91% |
> > | LSAT | 1.82% | 2.32% |
> > | AIME | 60.69% | 53.28% |
> > | MATH-500 | 1.31% | 1.33% |
> > | FRAMES | 13.11% | 11.37% |
> >
> >
> > Table 5: Routing performance on OOD queries across benchmarks reported on the hypervolume metric (higher is better).
> > | Benchmark (OOD) | RADAR | RADAR (MLP) |
> > | - | - | - |
> > | GPQA-Diamond | 0.7466 | 0.7245 |
> > | MMLU | 0.8609 | 0.8573 |
> > | MMLU-Redux | 0.9072 | 0.9067 |
> > | MMLU-Pro | 0.7858 | 0.7844 |
> > | LSAT | 0.9146 | 0.9101 |
> > | AIME | 0.7566 | 0.7726 |
> > | MATH-500 | 0.9368 | 0.9424 |
> > | FRAMES | 0.8865 | 0.8771 |
> >
> >
> > Table 6: Routing performance on OOD queries across benchmarks reported on the CPT (90%) metric (lower is better).
> > | Benchmark (OOD) | RADAR | RADAR (MLP) |
> > | - | - | - |
> > | GPQA-Diamond | 17.6% | 23.49% |
> > | MMLU | 2.63% | 2.82% |
> > | MMLU-Redux | 2.54% | 2.63% |
> > | MMLU-Pro | 3.54% | 4.82% |
> > | LSAT | 2.15% | 2.12% |
> > | AIME | 55.30% | 59% |
> > | MATH-500 | 1.55% | 1.46% |
> > | FRAMES | 9.99% | 11.17% |

---

> > > ### Author Response · Authors · 2025-11-25
> > > **Response to Reviewer EkVW (Part 3)**
> > >
> > > **Discussion of Inference-time Scaling Methods**
> > >
> > > Thank you for pointing us to these concurrent works on answer selection in multi-LLM and best-of-N settings [1,2]. Both methods assume a fixed set of models (or a single model) and generate multiple candidate responses per query, then use uncertainty or self-certainty scores to select the best answer at test time, effectively trading additional inference-time compute and latency for improved accuracy. In contrast, RADAR focuses on pre-inference routing: given a query, it selects a single model and reasoning length and then generates only one response, without requiring multiple samples or cross-model reruns. The routing overhead is negligible compared to a single LLM forward pass, so in its current instantiation our inference-time cost scales with one generation, while best-of-N answer selection scales with the number of samples. We view these approaches as complementary rather than competing: one could first use RADAR to choose a cost-effective model/CoT configuration and, when additional latency budget is available, apply uncertainty-aware or self-certainty-based answer selection on top. A systematic empirical study of such combinations and their latency–accuracy trade-offs is an interesting direction for future work but is beyond the scope of this paper.
> > >
> > >
> > >
> > > **Embedding Model**
> > >
> > > We used the Qwen3 Embedding model with 8B parameters which we report in Appendix D.3. This model is the best performing model on the MTEB multilingual leaderboard (as of June 5, 2025).
> > >
> > >
> > >
> > >
> > > **Dynamic Adjustment of Performance-Cost Tradeoff Weight**
> > >
> > > In our experiments we fix the weight $w_1$ for a given deployment scenario, but RADAR’s routing rule in Eq. (1) is defined for an arbitrary user-specified weight and does not require it to be static. In practice, one can plug in any policy $w_1(t)$ that depends on observed feedback or remaining budget, for example, treating $w_1$ as an online Lagrange multiplier [3,4] that is increased when cumulative cost is above the desired budget trajectory and decreased otherwise, so that later queries are routed to cheaper configurations. Implementing such a controller only changes how $w_1$ is chosen over time; the underlying per-query MOO and our IRT-based performance/cost predictors remain the same. Systematically designing and evaluating these adaptive policies is an interesting direction for future work and complementary to our current focus on the per-query routing formulation.
> > >
> > > [1] Agrawal A, Aralikatti R, Satheesh A, et al. Uncertainty-Aware Answer Selection for Improved Reasoning in Multi-LLM Systems[C]. Findings of the Association for Computational Linguistics: EMNLP 2025. 2025: 25090-25098.
> > >
> > > [2] Kang Z, Zhao X, Song D. Scalable best-of-n selection for large language models via self-certainty[J]. arXiv preprint arXiv:2502.18581, 2025.
> > >
> > > [3] Boyd S, Vandenberghe L. Convex optimization[M]. Cambridge university press, 2004.
> > >
> > > [4] Mahdavi M, Jin R, Yang T. Trading regret for efficiency: online convex optimization with long term constraints[J]. The Journal of Machine Learning Research, 2012, 13(1): 2503-2528.

---

### Official Review · Reviewer_aypQ · 2025-11-01

**Soundness:** 2
**Presentation:** 3
**Contribution:** 2
**Rating:** 4
**Confidence:** 3

**Summary:**

The authors proposed RADAR (Reasoning–Ability and Difficulty-Aware Routing), a lightweight, interpretable, and scalable routing framework. They firstly show with statistics that reasoning models today struggle to balance between performances and budgets. The authors then proposed to model this problem as a multi-objective optimization problem. With extensive evaluations on multiple benchmarks, the authors showed that their methods are superior to other methods, and the advantage could generalize to new RLM settings.

**Strengths:**

1. The paper is well written and nicely presented.
2. The method the authors proposed is simple, yet more reasonable to generalize.

**Weaknesses:**

1. The authors did not well explain their experimental results. For example, the gain looks marginal on some benchmarks while significant on others. The authors could help readers better understand why the biases exist.

**Questions:**

1.As I said above, could you give an explanation of your results?
2. The cost function looks simple. For example, how can RADAR incorporate real-time latency, KV-cache reuse, or batching discounts into its cost function?

---

> ### Author Response · Authors · 2025-11-25
> **Response to Reviewer aypQ (Part 1)**
>
> Thank you for your thorough review and insightful questions, and for highlighting the novelty and effectiveness of our method.
>
>
> **Additional Explanations of Results**
>
> In the ID setting, we perform marginally better or worse on just three datasets: AIME, MATH, and LSAT, and much better on the remaining datasets. We see a similar trend in the OOD setting with a slight decrease on AIME.
>
> We focus on the math-based datasets as representative examples. We compute the average query difficulty predicted by the IRT model in RADAR. We observe two extremes: AIME queries have the highest average query difficulty across benchmarks and MATH has the lowest query difficulty across benchmarks. These query difficulty predictions also correlate with our preliminary experiments in which we find performance on AIME queries consistently increases with an increase in model size and reasoning budget, while over 50% of MATH queries can be solved using the smallest RLM with a minimal reasoning budget (see Figure 7 in Appendix).
>
> The performance-cost landscape of AIME queries does not provide a suitable reasoning budget differential with RLMs at maximum budget (as used in IRT-Router) performing better. Further, the saturation of the MATH benchmark only allows for marginal gains among routing methods.
>
>
> Table 1: Predicted query difficulty on test questions. IRT difficulty values usually lie in [-3, 3] with a higher value denoting higher difficulty.
> | Benchmark | Diff Avg | Diff Std |
> | - | - | - |
> | AIME | 1.038 | 1.275 |
> | MATH-500 | -2.011 | 1.784 |
> | All Benchmarks | -0.55 | 1.018 |
>
>
>
> **Improving AIME OOD Performance**
>
> RADAR exhibits strong generalization performance to OOD queries consistently across benchmarks with a small decrease in performance compared with in-distribution queries even on long-context docQA. The only benchmark with a slight decrease in performance is AIME. Our analysis reveals two special characteristics of AIME. First, AIME contains queries with the highest average difficulty, significantly higher than other benchmarks. Second, performance on AIME consistently improves with an increase in reasoning length and does not plateau unlike other benchmarks. In the OOD setting, RADAR which is not exposed to difficult math problems, underestimates query difficulty and underperforms by allocating less capable models.
>
> Table 2: Predicted query difficulty on test questions. IRT difficulty values usually lie in [-3, 3] with a higher value denoting higher difficulty.
> | Benchmark | Diff Avg | Diff Std |
> | - | - | - |
> | AIME (ID) | 1.038 | 1.275 |
> | AIME (OOD)| -0.461 | 1.043 |
>
> For real-world usage, it is practical to assume some model training exposure to math problems. We perform a partial OOD experiment where we expose RADAR to a fraction of AIME and MATH queries in the training data. We see a significant increase in performance with just 5% of exposure, and a similar performance to the ID setting with 30% of exposure.
>
>
> Table 2: Partial OOD experimental setting showing a small set of training queries can recover ID performance.
> | Benchmark | RADAR (ID) | RADAR (OOD) | RADAR (OOD + 5%) | RADAR (OOD + 30%) |
> | - | - | - | - | - |
> | AIME | 0.776 | 0.7566 | 0.7665 | 0.7787 |

---

> > ### Author Response · Authors · 2025-11-25
> > **Response to Reviewer aypQ (Part 2)**
> >
> > **Cost Function Extensions**
> >
> > We agree that our current cost model is simple: in Sec. 3.4 we only account for dollar cost induced by reasoning tokens, as this is the dominant factor in our single-query setting and lets us isolate the effect of routing.
> >
> > However, the MOO formulation in Sec. 3.2 is agnostic to the particular definition of the cost term. In practice, $c_q(g)$ can be replaced by any learned or analytic estimate that combines multiple factors such as real-time latency, GPU memory, or KV-cache reuse, and amortized dollar cost under a batching policy. For example, one can normalize each of these quantities and either (i) combine them into a single scalar cost via a weighted (or Chebyshev-style) scalarization, or (ii) treat them as additional objectives in a higher-dimensional MOO [1,2], for which our Chebyshev scalarization directly applies. Batching discounts mainly change how cost is computed (e.g., as a function of current batch size and cache hit-rate); our present experiments consider one-query-at-a-time routing, but the optimization framework extends naturally to these richer, deployment-specific cost models, which we view as complementary future work.
> >
> > More generally, batched routing can be formulated as a joint assignment problem [3,4] over all queries in the batch, where the objective combines aggregate performance and a cost model that depends on batch composition (e.g., shared latency, KV-cache reuse, amortized dollar cost). Our per-query router is then a natural greedy baseline for this setting, and an interesting direction is to design approximate solvers or learned batch-level controllers that better exploit these batching effects while preserving RADAR’s multi-objective formulation.
> >
> > [1] Miettinen K. Nonlinear multiobjective optimization[M]. Springer Science & Business Media, 1999.
> >
> > [2] Ehrgott M. Multicriteria optimization[M]. Berlin, Heidelberg: Springer Berlin Heidelberg, 2005.
> >
> > [3] Kuhn H W. The Hungarian method for the assignment problem[J]. Naval research logistics quarterly, 1955, 2(1‐2): 83-97.
> >
> > [4] Burkard R, Dell'Amico M, Martello S. Assignment problems: revised reprint[M]. Society for Industrial and Applied Mathematics, 2012.

---

### Official Review · Reviewer_St66 · 2025-11-01

**Soundness:** 2
**Presentation:** 2
**Contribution:** 2
**Rating:** 4
**Confidence:** 4

**Summary:**

This paper presents a method for the reasoning language model (RLM) routing problem, which aims to select the best RLM configuration for an input query, given a pool of RLMs,  the reasoning budget, and a user performance-cost trade-off profile. It leverages an item response theory model to estimate the query difficulty and RLM ability at different reasoning budget, by parameterizing the query difficulty and training on collected evaluation responses. Under the user performance-cost trade-off profile, it uses multi-objective optimization to find the Pareto optimal of model configurations, which is also empirically shown to be effective.

**Strengths:**

- The problem setting of user-desired performance-cost trade-off profile is interesting and practical.
- The proposed approach of modeling query difficulty and solving the multi-objetive optimization problem is reasonable.
- RADAR can generalize to OOD queries.
- Comprehensive empirical evaluation on different benchmarks.

**Weaknesses:**

- Limited technical novelty against the baseline IRT-Router with conceptually similar approach of leveraging IRT.
- It is not convincing that using a linear transformation on the embedding of questions could successfully model the underlying difficulty of the question. Using a more expressive parameterization would  intuitively give a better results given the high dimensionality of the embedding. However, no ablation has been shown to justify the linear transformation is sufficient.
- The empirical gains are limited based on the results from Table 1, 2. The evaluation method CPT with only one threshold (i.e., CPT(90%)) is not convincing. More evaluation threshold should be shown to justify its effectiveness under different desirability.
- Typos. E.g., there is a missing “running” before “configuration” in line 196.
- Sec. 3.5 depends on a small set of evaluation queries, which can be adaptively updated. However, whether this set (queries with high Fisher information) approximates the global data distribution Q is unclear. Ablations on how this approach differs from uniform sampling or simply estimation from historical observations would be helpful.

**Questions:**

See weaknesses  above.

---

> ### Author Response · Authors · 2025-11-25
> **Response to Reviewer St66 (Part 1)**
>
> Thank you for your thorough review with insightful questions, and for highlighting the novelty and effectiveness of our method.
>
>
>
> **Comparison with IRT-Router**
>
> Compared to IRT-Router, a concurrent and recently released work, RADAR makes the following contributions:
>
> 1. Novel MOO Formulation: We're the first to formulate model routing in a mathematically principled way as multi-objective optimization (MOO) that searches for the model at the Pareto frontier of the performance-cost tradeoff curve using scalarization techniques. The objective used in IRT-Router (Eq 1) is introduced in an ad-hoc manner without justification and is a *special case* of our MOO formulation with linear scalarization. Beyond simple linear scalarization which cannot recover non-convex Pareto fronts, our MOO formulation allows the LLM routing community to leverage powerful solution techniques from well-established MOO literature including Chebyshev scalarization. In RADAR, we find that Chebyshev scalarization outperforms linear scalarization on the stable hypervolume metric, in the challenging OOD experimental setting, due to its ability to explore both convex and concave points on the Pareto front. In the easier ID experimental setting, both scalarization techniques perform similarly, with linear being marginally better. We leave exploration of other MOO solution techniques like lexicographic methods for future work. We show results on the hypervolume metric on linear vs Chebyshev scalarization on OOD benchmarks below with complete results included in Table 5 in Appendix E.
>
>
> Table 1: Ablation study showing Chebyshev scalarization outperforms linear scalarization  on OOD queries due to its ability to explore both convex and concave points on the Pareto front.
> | Benchmark (OOD) | RADAR (Linear) | RADAR (Chebyshev) |
> | - | - | - |
> | GPQA-Diamond | 0.7280 | 0.7466 |
> | MMLU | 0.8580 | 0.8609  |
> | MMLU-Redux | 0.9049 | 0.9072 |
> | MMLU-Pro | 0.7812 | 0.7858  |
> | LSAT | 0.9165 | 0.9146  |
> | AIME | 0.7464 | 0.7566  |
> | MATH-500 | 0.9331 | 0.9368  |
> | FRAMES | 0.8656 | 0.8865  |
>
> 2. Model Generalizability: Another significant contribution of RADAR is its effective model generalization capability. IRT-Router uses an ad-hoc approach of querying ChatGPT (web search mode) for a description of the new LLM which is further corrected by *manual intervention* and then embedded. IRT-Router notes this limited generalization of their approach and highlights model generalizability as an important direction of future work. In contrast, to add a new RLM configuration in RADAR we simply need its ability. To precisely estimate the ability of a new RLM configuration, RADAR follows a *principled and fully automated* approach by evaluating it on a small set (*12% of training queries*) of dynamically selected queries, employing a classic technique inspired by adaptive testing in educational assessment.
> 3. Informative Metrics: IRT-Router reports performance at three arbitrary performance-cost tradeoff weights (0.2, 0.5, and 0.8) which fails to provide an accurate evaluation of routing methods. In contrast, our MOO formulation of routing allows us to leverage the hypervolume metric from MOO, which corresponds to the area under the performance-cost trade-off curve across the entire domain (from 0 to 1) of the trade-off weight, leading to a more stable and informative metric.
> 4. Custom Interpretable IRT Model: IRT-Router employs a standard Multidimensional-IRT (MIRT) model which uses non-interpretable vectorized abilities. In contrast, RADAR uses a custom, interpretable IRT model with multidimensional embeddings for queries to allow for OOD generalizability and scalar model abilities to allow for *interpretable* ability ordering among models. This ability ordering, as seen in the y-axis of Figure 3, is useful for RLM benchmarking and evaluation.
> 5. Focus on Reasoning LLMs: In contrast to past work including IRT-Router which are primarily focused on LLMs, we present a routing formulation for RLMs incorporating reasoning budgets. Choosing the right RLM for practical deployment involves a performance and cost trade-off at two key levels: base models and reasoning budgets. We *unify* these decisions by *discretizing* each RLM by its available set of reasoning budgets. Although simple, our discretization trick easily extends to select among other model settings, such as parameterizations of the RAG pipeline attached to the RLM or decoding methods employed.

---

> > ### Author Response · Authors · 2025-11-25
> > **Response to Reviewer St66 (Part 2)**
> >
> > **Ablation Study on Linear Transform for Query Difficulty**
> >
> > Our design choice of simple, interpretable linear transformations and a frozen embedding model helps ensure a minimal router latency of around 7 milliseconds. We agree that more complicated architectures could better model the difficulty of queries. However, our goal in RADAR is not to find the best architecture but to present an easily customizable MOO-grounded routing framework with design choices including base IRT models, transformations for difficulty, MOO solution techniques, among others.
> >
> > We test the sufficiency of our linear transformation for query difficulty in two ways. First, we performed a case study on MATH-500 where queries are annotated with one of five levels of increasing difficulty, one of the few datasets with real problem difficulty annotations. We find a moderately positive Pearson correlation coefficient of 0.509 between RADAR-estimated query difficulties and the five ground-truth levels, showing that RADAR is able to capture real question difficulty. We have reported our findings in Section 4.3.
> >
> > Second, we perform a new ablation study with a classic two-layer multilayer perceptron (MLP) model using a ReLU non-linearity to obtain the difficulty and discrimination of queries. Our results on the hypervolume metric show similar performance in both the ID and OOD experimental setting across benchmarks. On AIME and MATH-500, MLP is slightly worse in the ID setting, and sbetter in the OOD setting. These none to marginal gains, obtained at the cost of diminished interpretability and increased latency, further justify the simplicity of our design choices in RADAR.
> >
> >
> > Table 2: Routing performance on ID queries across benchmarks reported on the hypervolume metric (higher is better).
> > | Benchmark (ID) | RADAR | RADAR (MLP) |
> > | - | - | - |
> > | GPQA-Diamond | 0.7513 | 0.7308 |
> > | MMLU | 0.8720 | 0.8707 |
> > | MMLU-Redux | 0.9230 | 0.9239 |
> > | MMLU-Pro | 0.7995 | 0.7955 |
> > | LSAT | 0.9188 | 0.9132 |
> > | AIME | 0.7760 | 0.7687 |
> > | MATH-500 | 0.9449 | 0.9365 |
> > | FRAMES | 0.8762 | 0.8777 |
> >
> >
> > Table 3: Routing performance on ID queries across benchmarks reported on the CPT (90%) metric (lower is better).
> > | Benchmark (ID) | RADAR | RADAR (MLP) |
> > | - | - | - |
> > | GPQA-Diamond | 13.21% | 14.11% |
> > | MMLU | 2.69% | 2.8% |
> > | MMLU-Redux | 2.42% | 2.51% |
> > | MMLU-Pro | 3.89% | 3.91% |
> > | LSAT | 1.82% | 2.32% |
> > | AIME | 60.69% | 53.28% |
> > | MATH-500 | 1.31% | 1.33% |
> > | FRAMES | 13.11% | 11.37% |
> >
> >
> > Table 4: Routing performance on OOD queries across benchmarks reported on the hypervolume metric (higher is better).
> > | Benchmark (OOD) | RADAR | RADAR (MLP) |
> > | - | - | - |
> > | GPQA-Diamond | 0.7466 | 0.7245 |
> > | MMLU | 0.8609 | 0.8573 |
> > | MMLU-Redux | 0.9072 | 0.9067 |
> > | MMLU-Pro | 0.7858 | 0.7844 |
> > | LSAT | 0.9146 | 0.9101 |
> > | AIME | 0.7566 | 0.7726 |
> > | MATH-500 | 0.9368 | 0.9424 |
> > | FRAMES | 0.8865 | 0.8771 |
> >
> >
> > Table 5: Routing performance on OOD queries across benchmarks reported on the CPT (90%) metric (lower is better).
> > | Benchmark (OOD) | RADAR | RADAR (MLP) |
> > | - | - | - |
> > | GPQA-Diamond | 17.6% | 23.49% |
> > | MMLU | 2.63% | 2.82% |
> > | MMLU-Redux | 2.54% | 2.63% |
> > | MMLU-Pro | 3.54% | 4.82% |
> > | LSAT | 2.15% | 2.12% |
> > | AIME | 55.30% | 59% |
> > | MATH-500 | 1.55% | 1.46% |
> > | FRAMES | 9.99% | 11.17% |

---

> > > ### Author Response · Authors · 2025-11-25
> > > **Response to Reviewer St66 (Part 3)**
> > >
> > > **Empirical Gains and CPT Metric on Additional Threshols**
> > >
> > > The hypervolume metric is more stable and is agnostic to the choice of the performance-cost tradeoff weight. On hypervolume in the ID setting, RADAR perform marginally better or worse on just three datasets: AIME, MATH, and LSAT, and much better on all remaining datasets. We see a similar trend in the OOD setting with a slight decrease on AIME due to its significantly higher average query difficulty requiring larger models with larger reasoning budgets which provides an unsuitable performance differential for routing.
> > >
> > >
> > > We report CPT(85%) below and have added additional thresholds (80%, 85%, 90%, and 95%) in Appendix E.5. On CPT(80%), simpler models like RouterBench perform better on some benchmarks possibly due to the lower threshold not being discriminative enough. We find similar trends in results as shown previously on CPT(90%). We note that the CPT metric is a high variance performance estimate and results on the hypervolume metric which is agnostic to the performance-cost tradeoff weight are moreins reliable.
> > >
> > >
> > >
> > > Table 6: Routing performance on ID queries across benchmarks reported on the CPT(85%) metric (lower is better). CPT(85%) denotes the fraction of cost of running OpenAI o4-mini with high reasoning effort to match 85% of its performance.
> > > | Benchmark | Random-Pair | RouterBench | IRT-Router | RADAR (ours) |
> > > | --- | --- | --- | --- | --- |
> > > | GPQA-Diamond | $71.53\%$ | $37.41\%$ | $\underline{31.19\%}$ | $\mathbf{8.93\%}$ |
> > > | MMLU | $64.46\%$ | $\mathbf{1.61\%}$ | $1.85\%$ | $\underline{1.68\%}$ |
> > > | MMLU-Redux | $63.77\%$ | $\underline{1.85\%}$ | $1.97\%$ | $\mathbf{1.67\%}$ |
> > > | MMLU-Pro | $75.91\%$ | $\underline{2.18\%}$ | $\mathbf{2.14\%}$ | $2.23\%$ |
> > > | LSAT | $70.35\%$ | $1.73\%$ | $\underline{1.64\%}$ | $\mathbf{1.33\%}$ |
> > > | AIME | $80.84\%$ | $45.19\%$ | $\underline{42.39\%}$ | $\mathbf{41.04\%}$ |
> > > | MATH-500 | $64.97\%$ | $\underline{0.94\%}$ | $0.95\%$ | $\mathbf{0.93\%}$ |
> > > | FRAMES | $69.41\%$ | $10.17\%$ | $\underline{1.22\%}$ | $\mathbf{1.21\%}$ |
> > >
> > >
> > >
> > > Table 7: Routing performance on OOD queries across benchmarks reported on the CPT(85%) metric (lower is better). CPT(85%) denotes the fraction of cost of running OpenAI o4-mini with high reasoning effort to match 85% of its performance.
> > > | Benchmark | Random-Pair | RouterBench | IRT-Router | RADAR (ours) |
> > > | --- | --- | --- | --- | --- |
> > > | GPQA-Diamond | $74.3\%$ | $\underline{24.53\%}$ | $31.34\%$ | $\mathbf{10.86\%}$ |
> > > | MMLU | $63.55\%$ | $1.96\%$ | $\underline{1.88\%}$ | $\mathbf{1.73\%}$ |
> > > | MMLU-Redux | $63.09\%$ | $1.99\%$ | $\underline{1.92\%}$ | $\mathbf{1.85\%}$ |
> > > | MMLU-Pro | $74.4\%$ | $2.44\%$ | $\underline{2.2\%}$ | $\mathbf{2.13\%}$ |
> > > | LSAT | $70.54\%$ | $1.73\%$ | $\underline{1.65\%}$ | $\mathbf{1.5\%}$ |
> > > | AIME | $77.92\%$ | $100.0\%$ | $\mathbf{32.78\%}$ | $\underline{42.9\%}$ |
> > > | MATH-500 | $63.55\%$ | $\underline{0.9\%}$ | $0.97\%$ | $\mathbf{0.73\%}$ |
> > > | FRAMES | $68.74\%$ | $16.7\%$ | $\mathbf{1.21\%}$ | $\underline{1.55\%}$ |
> > >
> > >
> > >
> > > **Fixed Typos**
> > >
> > > Thank you for pointing these out. We've fixed them.

---

> > > > ### Author Response · Authors · 2025-11-25
> > > > **Response to Reviewer St66 (Part 4)**
> > > >
> > > > **Evaluation and Ablations on Adaptive Testing**
> > > >
> > > > Thank you for your excellent suggestion. We perform an ablation study on our Fisher information-based adaptive testing compared with uniform random sampling on routing performance. Similar to well-established findings in adaptive testing [1,2] and online learning [3,4,5,6] literature, we find that our method performs better and with lower variance than uniform sampling especially when sample size is low, and both methods converge to perform similarly as sample size increases.
> > > >
> > > > For example, with a small sample size of just 100 items, our method consistently outperforms uniform sampling by a wide margin on ID benchmarks and on AIME and MATH on OOD benchmarks. Further, uniform sampling exhibits high variance as seen in FRAMES in the ID setting achieving 2.6% CPT(90%) with 100 items which counterintuitively jumps to 22.2% with a larger set of 500 items. When uniform sampling marginally performs better, its high variance might indicate a fortuitous estimate of ability rather than an accurate one. In contrast, our method provides a reliable and precise estimate of ability which results in stable routing performance across sample sizes.
> > > >
> > > > Table 8: Routing performance on ID queries across benchmarks reported on the hypervolume metric (higher is better), before (RADAR) and after (RADAR++) adding new RLM configurations from Qwen3-14B. Random X denotes set of X items selected with uniform random sampling while Fisher X denotes set of X items selected with our method of maximum Fisher information-based adaptive testing. RADAR++
> > > > | Benchmark (ID) | RADAR | RADAR++ (Random 100) | RADAR++ (Fisher 100) | RADAR++ (Random 500) | RADAR++ (Fisher 500) | RADAR++ (Random 5000) | RADAR++ (Fisher 5000) |
> > > > | - | - | - | - | - | - | - | - |
> > > > | GPQA-Diamond | 0.7513 | 0.6171 | 0.7511 | 0.7017 | 0.7513 |  0.7535  | 0.7535 |
> > > > | MMLU | 0.8609 | 0.8591 | 0.8731 | 0.8682 | 0.8720 | 0.8745 | 0.8731 |
> > > > | MMLU-Redux | 0.9072 | 0.8929 | 0.9232 | 0.9168 | 0.9230 | 0.9228 | 0.9238 |
> > > > | MMLU-Pro | 0.7858 | 0.7764 | 0.8009 | 0.7941 | 0.7995 | 0.8038 | 0.8021 |
> > > > | LSAT | 0.9146 | 0.9270 | 0.9205 | 0.9283 | 0.9188 | 0.9259 | 0.9233 |
> > > > | AIME | 0.7760 | 0.6883 | 0.7636 | 0.7311 | 0.7760 | 0.7840 | 0.7828 |
> > > > | MATH-500 | 0.9449 | 0.9426 | 0.9456 | 0.9457 | 0.9449 | 0.9478 | 0.9461 |
> > > > | FRAMES | 0.8865 | 0.8629 | 0.8763 | 0.8587 | 0.8762 | 0.8897 | 0.8830 |
> > > >
> > > >
> > > > Table 9: Routing performance on ID queries across benchmarks reported on the CPT (90%) metric (lower is better), before (RADAR) and after (RADAR++) adding new RLM configurations from Qwen3-14B. Random X denotes set of X items selected with uniform random sampling while Fisher X denotes set of X items selected with our method of maximum Fisher information-based adaptive testing. NR denotes not reachable.
> > > > | Benchmark (ID) | RADAR | RADAR++ (Random 100) | RADAR++ (Fisher 100) | RADAR++ (Random 500) | RADAR++ (Fisher 500) | RADAR++ (Random 5000) | RADAR++ (Fisher 5000) |
> > > > | - | - | - | - | - | - | - | - |
> > > > | GPQA-Diamond | 13.21% | NR | 13.61% | 45.24% | 13.21% | 12.28% | 13.56% |
> > > > | MMLU | 2.63% | 2.67% | 2.69% | 2.75% | 2.69% | 2.69% | 2.69% |
> > > > | MMLU-Redux | 2.54% | 2.38% | 2.42% | 2.42% | 2.42% | 2.42% | 2.42% |
> > > > | MMLU-Pro | 3.54% | 3.73% | 3.9% | 3.93% | 3.89% | 3.9% | 3.9% |
> > > > | LSAT | 2.15% | 1.82% | 1.82% | 1.82% | 1.82% | 1.82% | 1.82% |
> > > > | AIME | 60.69% | NR | 65.12% | 72.43% | 60.69% | 57.7% | 58.45% |
> > > > | MATH-500 | 1.31% | 1.34% | 1.31% | 1.32% | 1.31% | 1.32% | 1.31% |
> > > > | FRAMES | 9.99% | 2.62% | 13.06% | 22.20% | 13.11% | 3.25% | 5.97% |

---

> > > > > ### Author Response · Authors · 2025-11-25
> > > > > **Response to Reviewer St66 (Part 5)**
> > > > >
> > > > > Table 10: Routing performance on OOD queries across benchmarks reported on the hypervolume metric (higher is better), before (RADAR) and after (RADAR++) adding new RLM configurations from Qwen3-14B. Random X denotes set of X items selected with uniform random sampling while Fisher X denotes set of X items selected with our method of maximum Fisher information-based adaptive testing.
> > > > > | Benchmark (OOD) | RADAR | RADAR++ (Random 100) | RADAR++ (Fisher 100) | RADAR++ (Random 500) | RADAR++ (Fisher 500) | RADAR++ (Random 5000) | RADAR++ (Fisher 5000) |
> > > > > | - | - | - | - | - | - | - | - |
> > > > > | GPQA-Diamond | 0.7466 | 0.7402 | 0.6362 | 0.7466 | 0.7177 | 0.7467 | 0.7463 |
> > > > > | MMLU | 0.8609 | 0.87 | 0.8679 | 0.8697 | 0.8684 | 0.8697 | 0.8698 |
> > > > > | MMLU-Redux | 0.9072 | 0.9086 | 0.9093 | 0.9090 | 0.9098 | 0.9094 | 0.9091 |
> > > > > | MMLU-Pro | 0.7858 | 0.7966 | 0.7927 | 0.7958 | 0.7928 | 0.7961 | X |
> > > > > | LSAT | 0.9146 | 0.9287 | 0.9273 | 0.9264 | 0.9267 | 0.9278 | 0.9255 |
> > > > > | AIME | 0.7566 | 0.6543 | 0.7566 | 0.7385 | 0.7566 | 0.7694 | 0.7566 |
> > > > > | MATH-500 | 0.9368 | 0.9186 | 0.9368 | 0.9367 | 0.9368 | 0.9389 | 0.9368 |
> > > > > | FRAMES | 0.8865 | 0.8658 | 0.8699 | 0.8864 | 0.8872 | 0.8938 | 0.8931 |
> > > > >
> > > > >
> > > > > Table 11: Routing performance on OOD queries across benchmarks reported on the CPT (90%) metric (lower is better), before (RADAR) and after (RADAR++) adding new RLM configurations from Qwen3-14B. Random X denotes set of X items selected with uniform random sampling while Fisher X denotes set of X items selected with our method of maximum Fisher information-based adaptive testing. NR denotes not reachable.
> > > > > | Benchmark (OOD) | RADAR | RADAR++ (Random 100) | RADAR++ (Fisher 100) | RADAR++ (Random 500) | RADAR++ (Fisher 500) | RADAR++ (Random 5000) | RADAR++ (Fisher 5000) |
> > > > > | - | - | - | - | - | - | - | - |
> > > > > | GPQA-Diamond | 17.6% | 22.6% | NR | 15.8% | 37% | 17.48% | 16.63% |
> > > > > | MMLU | 2.63% | 2.63% | 2.63% | 2.63% | 2.63% | 2.63% | 2.63% |
> > > > > | MMLU-Redux | 2.54% | 2.54% | 2.54% | 2.54% | 2.54% | 2.54% | 2.54% |
> > > > > | MMLU-Pro | 3.54% | 3.69% | 3.54% | 3.69% | 3.54% | 3.7% | 3.56% |
> > > > > | LSAT | 2.15% | 2.19% | 2.15% | 2.53% | 2.15% | 2.48% | 2.15% |
> > > > > | AIME | 55.3% | 59.2% | 55.3% | 56.5% | 55.3% | 52.5% | 55.3% |
> > > > > | MATH-500 | 1.55% | 1.98% | 1.55% | 1.53% | 1.55% | 1.57% | 1.55% |
> > > > > | FRAMES | 9.99% | 8.72% | 2.58% | 10% | 5.16% | 2.57% | 5.25% |
> > > > >
> > > > >
> > > > > [1] Kürşad M Ş. The Effects of Different Item Selection Methods on Test Information and Test Efficiency in Computer Adaptive Testing[J]. Journal of Measurement and Evaluation in Education and Psychology, 2023, 14(1): 33-46.
> > > > >
> > > > > [2] Dwahdh L, Alshraifin N. The impact of computerized adaptive test termination rules on accuracy across different ability estimation methods[J]. Eurasia Journal of Mathematics, Science and Technology Education, 2025, 21(1): em2571.
> > > > >
> > > > > [3] Hanneke S. Theory of active learning[J]. Foundations and Trends in Machine Learning, 2014, 7(2-3).
> > > > >
> > > > > [4] Balcan M F, Hanneke S, Vaughan J W. The true sample complexity of active learning[J]. Machine learning, 2010, 80(2): 111-139.
> > > > >
> > > > > [5] Hacohen G, Weinshall D. How to select which active learning strategy is best suited for your specific problem and budget[J]. Advances in Neural Information Processing Systems, 2023, 36: 13395-13407.
> > > > >
> > > > > [6] Cacciarelli D, Kulahci M. Active learning for data streams: a survey[J]. Machine Learning, 2024, 113(1): 185-239.

---

### Author Response · Authors · 2025-12-03
**Summary of Changes Made in Revised Manuscript**

We sincerely thank the reviewers (R1:St66, R2:aypQ, R3:EkVW, R4:LEBC) for their helpful suggestions and for highlighting the novelty and effectiveness of our method. We have uploaded a revised manuscript that addresses their feedback, with changes highlighted in blue.



We thank the reviewers for highlighting the following contributions and strengths of our paper, summarized below:

1. Clear motivation with interesting and practical problem setting (R1, R4).
2. First to formulate model routing as a multi-objective optimization (MOO) problem (R3).
3. Rigorous methodology offering a generalizable and practical approach (R2, R3).
4. Fast, interpretable approach with low latency (R4).
5. Innovative use of IRT for modeling reasoning ability and query difficulty (R3).
6. Comprehensive empirical evaluation on 8 different benchmarks in both ID and OOD settings (R1, R3).
7. RADAR can generalize to OOD queries (R1).
8. The paper is very well written and structured with clear motivation (R2, R3).



We summarize the key changes made addressing reviewer feedback below:

1. **Ablation Study on Linear Transform for Query Difficulty (R1, R3, R4)**: Our design choice of simple linear transformations for query difficulty makes RADAR interpretable with low latency. We've added an ablation study against a classic two-layer MLP with non-linearity in Appendix E.5, in which we find none to marginal gains using a more complex model. We conducted a case study on MATH-500, a dataset with real problem-difficulty annotations. We observe a moderate positive Pearson correlation coefficient of 0.509 between RADAR-estimated difficulties and ground-truth difficulties. We have already reported these findings in Section 4.3.
2. **Ablation on Adaptive Testing (R1)**: We added an ablation study on our Fisher information-based adaptive testing compared with uniform random sampling in Appendix E.6. Following findings from online learning, we find that our method performs better and with lower variance than uniform sampling especially when sample size is low (even with just 100 queries), and that both methods converge to perform similarly as sample size increases.
3. **Improving AIME OOD Performance (R3)**: In the OOD setting, removing all math-based datasets, RADAR underestimates AIME query difficulties, which have an exceptionally high difficulty. In real-world use, it is practical to assume some exposure to math problems during model training. We've added results from a partial OOD experiment in Appendix E.8, showing a significant increase in performance with just 5% exposure to AIME training queries, and a similar performance to the ID setting with 30% exposure.
4. **Comparison with IRT-Router (R1)**: We've added additional clarification in Appendix B to highlight the novelty of RADAR including a) its novel multi-objective optimization (MOO) formulation, b) better exploration of the Pareto frontier using Chebyshev scalarization (see Appendix E.2), c) fully automated addition of new models, d) custom, interpretable IRT model, e) informative metrics, and f) evaluation on reasoning models and challenging benchmarks.
7. **Additional results on CPT metric (R1)**: We've added results on the CPT metric at various thresholds, including 80%, 85%, 90% and 95% in Appendix E.7. We observe similar patterns with RADAR outperforming baselines on a majority of the datasets in both ID and OOD experimental settings.
8. **Throughput analysis (R3)**: We added a throughput analysis in Appendix E.9. The negligible routing overhead of RADAR (~7 milliseconds) decreases throughput by less than 1%.

---

### Meta-Review · Area_Chair_NRUT · 2025-12-09

**Summary:**

Reviewers were split 1 positive and 3 negative, although the 3 negative reviewers noted that the paper had many strengths in presentation, formulation and rigor. Reviewers were concerned about insufficient model expressiveness for the difficulty of the problem, and the need for more experiments given some performance gaps that might be related to said insufficient model expressiveness.

**Reviewer Concerns:**

Insufficient model expressiveness
- Authors argued that the linear transformations are needed to maintain low latency, and provided additional experiments showing that additional complexity does not improve performance. I believe this issue has been sufficiently addressed.

Need for additional experiments
- Authors provided new abalation studies, as well as the abovementioned experiments on linear vs more complex transformations. I think this issue has been at least partially addressed, if not more.

**Reviewer Scores:**

LEBC: review was already very positive, and did not raise serious concerns. I don't believe their score would have changed.

Of the 3 negative reviewers, I think a majority of them would have changed their score to positive.

---

### Decision · Program_Chairs · 2026-01-26

Accept (Poster)